# Learning Task-Aware Abstract Representations for Meta-Reinforcement Learning

**Louk van Remmerden**                                    *l.smalbil@vu.nl*
*Department of Computer Science*
*Vrije Universiteit Amsterdam*

**Zhao Yang**                                             *z.yang3@vu.nl*
*Department of Computer Science*
*Vrije Universiteit Amsterdam*

**Shujian Yu**                                            *s.yu3@vu.nl*
*Department of Computer Science*
*Vrije Universiteit Amsterdam*

**Mark Hoogendoorn**                                      *m.hoogendoorn@vu.nl*
*Department of Computer Science*
*Vrije Universiteit Amsterdam*

**Vincent François-Lavet**                                *vincent.francoislavet@vu.nl*
*Department of Computer Science*
*Vrije Universiteit Amsterdam*

**Reviewed on OpenReview:** *https://openreview.net/pdf?id=3CWyTh4hJ4*

## Abstract

A central challenge in meta-reinforcement learning (meta-RL) is enabling agents trained on a set of environments to generalize to new, related tasks without requiring full policy retraining. Existing model-free approaches often rely on context-conditioned policies learned via encoder networks. However, these context encoders are prone to overfitting to the training environments, resulting in poor out-of-sample performance on unseen tasks. To address this issue, we adopt an alternative approach that uses an abstract representation model to learn augmented, task-aware abstract states. We achieve this by introducing a novel architecture that offers greater flexibility than existing recurrent network-based approaches. In addition, we optimize our model with multiple loss terms that encourage predictive, task-aware representations in the abstract state space. Our method simplifies the learning problem and provides a flexible framework that can be readily combined with any off-the-shelf reinforcement learning algorithm. We provide theoretical guarantees alongside empirical results, showing strong generalization performance across classical control and robotic meta-RL benchmarks, on par with state-of-the-art meta-RL methods and significantly better than non-meta RL approaches.

## 1 Introduction

Achieving robust generalization to new environments is challenging for agents trained within a single setting. While such agents may perform well in the training environment, they often fail when faced with even minor changes to dynamics. Meta-reinforcement learning (meta-RL) addresses this limitation by training across a distribution of tasks, thereby enabling the few-shot adaptation of classical RL agents (Beck et al., 2025; Nagabandi et al., 2018).

One of the most promising directions in meta-RL is context conditioning (Zintgraf et al., 2021; Lee et al., 2020). The core idea is to learn a context representation that captures task-specific information from a distribution of environments. At test time, the agent can explicitly infer the context of a new, unseen environment and adapt its behavior accordingly. Recent work (Lee et al., 2020; Rimon et al., 2024) has shown that context-conditioned dynamics models are particularly effective to meta-RL settings.

A key limitation of context-based meta-RL approaches is that the agent's performance depends heavily on the encoder's ability to accurately distinguish between tasks. Consequently, if the context encoder overfits to the training distribution - or fails to generalize - its out-of-sample performance can degrade substantially (Zintgraf et al., 2021).

This work aims to improve generalization in meta-RL by leveraging abstract representation models (ARMs), which learn high-level representations of environmental states using a pretrained model. Our goal is to produce task-aware, augmented abstract state representations. Such an augmented abstract state is a latent representation that may include additional task information, analogous to a belief or hyper-state. Accordingly, this differs from the more conventional usage in which abstraction typically implies a compressed state with reduced information.

We obtain task-aware[1] states with a neural architecture that encodes states into an abstract space through a state-space encoder and infers task encodings with a recurrent neural network (RNN). These representations are optimized jointly using a set of complementary loss functions. Our approach offers two main advantages: (1) ARMs are straightforward to train and generalize well, reducing reliance on the context encoder; and (2) they integrate seamlessly with any off-the-shelf model-free reinforcement-learning algorithm, giving the agent direct access to task information.

Our network architecture integrates abstract state representation learning with task inference in a unified framework. Unlike previous approaches that rely primarily on task-encoders that are trained separately, we introduce a joint optimization framework based on predictive losses in the abstract space—capturing transitions, rewards, and task structure.

We call our methodology **E**nvironment-aware **M**eta **E**ncoding and **R**epresentation **A**bstraction for **L**atent **D**omains—**EMERALD**. It is a novel reinforcement learning framework that learns *task-aware abstract representations* capturing both environmental dynamics and task-specific factors, enabling improved generalization and task disambiguation in meta-RL settings.[2] Our key contributions are:

- EMERALD's performance is on par with *state-of-the-art performance* meta-RL methods across a suite of meta-RL benchmarks and outperforms classical RL approaches in *few-shot adaptation and transfer* tasks, and is compatible with standard reinforcement learning algorithms such as SAC and PPO.

- We provide *theoretical justification* for EMERALD's design, showing how incorporating contextual information into the abstract state space promotes fast generalization to unseen tasks.

## 2 Related Work

**Meta Reinforcement Learning** Meta-reinforcement learning (meta-RL) enables RL agents to rapidly adapt to new, unseen tasks or environments (Beck et al., 2025; Nagabandi et al., 2018). The goal is to learn a policy trained on a set of tasks $\mathcal{T}_i \sim p(\mathcal{T}_{\text{train}})$ that can quickly adapt to new tasks. Achieving this requires the agent to generalize effectively across tasks, leveraging experience from the training distribution to perform well in novel settings (Finn et al., 2017). Meta-RL can also be viewed through the lens of partially observable Markov decision processes (POMDPs), where the true task identity is unobserved and must be inferred through interaction, making context inference analogous to belief state estimation (Humplik et al., 2019; Rakelly et al., 2019).

---

[1]N.B. Our use of "task-aware" follows the meta-RL convention, where the representation is conditioned on an inferred context vector. In the continual learning literature, the same term typically denotes access to an explicit task identity (and "task-agnostic" the absence thereof).

[2]Code base can be found at https://github.com/ljsmalbil/EMERALD.

**Context-based Meta-RL** Context-based meta-RL approaches learn a context encoder to capture variations across tasks. PEARL (Rakelly et al., 2019) and VariBAD (Zintgraf et al., 2021) formulate the context-encoding problem using a Bayesian Adaptive MDP (BAMDP), where a context variable $z$ is modeled as a belief state inferred from past transitions and used to condition the policy, yielding $\pi(s_t \mid z)$. VariBAD passes an augmented state—consisting of the current state and the belief variable—to the policy. This allows the agent to adapt its behavior based on the inferred task identity. We build on this idea of an augmented state space, but rather than concatenating the state and context, we feed the learned context variable directly into the state-space encoder of the representation model.

**Model-based Contextual Meta-RL** MAMBA (Rimon et al., 2024) extends VariBAD to the model-based setting by integrating context encoding into Dreamer's latent imagination process. A related method, CaDM (Lee et al., 2020), learns contextualized dynamics models by conditioning both the reward and transition models on context, and employs a backward-and-forward transition mechanism to stabilize training. Our approach builds on these methods, but uses only forward predictions in the abstract space. As in Dreamer-based methods such as MAMBA, we allow the policy to operate in an abstract space with lower cardinality than the raw state space, $|\mathcal{X}| \ll |\mathcal{S} \times \mathcal{T}|$. However, our state–task encoder can also be deployed in model-free settings via abstract representation models. Figure 1 illustrates the high-level architectural differences between our approach and other context-based methods.

**Abstract Representation Models in Deep RL** Abstract representation models (ARMs) are widely used in deep reinforcement learning to improve generalization and sample efficiency (Botteghi et al., 2022; Starre et al., 2022; Ni et al., 2024). These models map high-dimensional states $\mathcal{S}$ to compact latent spaces $\mathcal{X}$ via encoders $\psi : \mathcal{S} \to \mathcal{X}$, enabling more structured learning. Unlike "standard" auto-encoders, abstract representation models in the context of RL are typically used to obtain generalizable abstractions of the reward and transition dynamics. Several works illustrate this: for instance, Zhang et al. (2018) decouple dynamics and rewards for domain generalization, while Li et al. (2021) apply abstract models to adversarial settings with shared dynamics but varying visual observations. We build on these insights and extend ARMs to the meta-RL setting. Similar to prior combined RL approaches (François-Lavet et al., 2019; Lee et al., 2020), our method is flexible and can be integrated with both model-free and model-based agents.

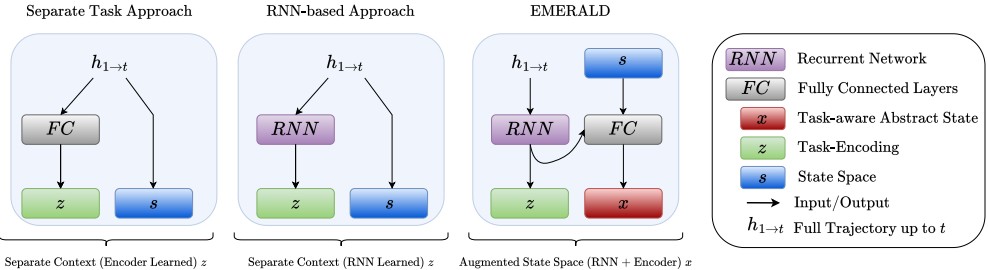

Figure 1: Comparison of state and context representation strategies. **Left:** Methods such as CaDM learn state ($s$) and context ($z$) representations independently, without temporal modeling. **Middle:** VariBAD-like approaches use RNNs to learn $z$ from trajectory history $h_{1 \to t}$, but still treat $s$ and $z$ as separate components. **Right:** Our proposed EMERALD ARM architecture integrates the trajectory-derived context $z$ with the current abstract state to form a task-aware shared latent state representation $x$, enabling unified and task-aware modeling of the environment dynamics.

## 3 Task-aware Abstract Representation Models

**Notation** We define a task $\mathcal{T}_i$ as a Markov Decision Process (MDP), represented by the 5-tuple $\mathcal{T}_i = (\mathcal{S}, \mathcal{A}, P_i, r_i, \gamma)$, where $\mathcal{S}$ is the state space, $\mathcal{A}$ the action space, $P_i(s_{t+1} \mid s_t, a_t)$ the transition kernel, and $r_i$ the reward function mapping state-action pairs to real-valued rewards. The discount factor is denoted $\gamma \in [0, 1)$. We index time steps by the subscript $t$, e.g., $s_t$ and $s_{t+1}$.

We sample $N$ independent and identically distributed (i.i.d.) training tasks $\mathcal{T}_i \sim p(\mathcal{T})$. From a task, we can collect data sets of transitions $\mathcal{D}_i = \{(s_{k,t}, a_{k,t}, s_{k,t+1}, r_{k,t})\}_{k=1}^{m_i}$. The combined dataset across all $N$ tasks is denoted $\mathcal{D} = \bigcup_{i=1}^{N} \mathcal{D}_i$. We let $m_i = |\mathcal{D}_i|$ denote the number of transitions collected from task $\mathcal{T}_i$. We denote the entropy of a random variable $X$ as $\mathbb{H}(X)$, and the conditional entropy as $\mathbb{H}(X \mid Z)$. $\mathcal{L}_i(\theta)$ denotes the loss of a model with parameters $\theta$ in a single environment $i$ and $\mathbb{E}_{\mathcal{T}_i \sim p(\mathcal{T})} \mathcal{L}_{\mathcal{T}}(\theta)$ the loss over all tasks in the distribution.

**Model Components** The abstract *state* space is denoted as $\mathcal{X}$ and the *task* embedding space as $\mathcal{Z}$. Note that, in our architecture, $\mathcal{X}$ contains both the information about the task as well as the state in that task. The ARM has four components: (1) the contextual state-space encoder $\psi : \mathcal{S} \times \mathcal{Z} \to \mathcal{X}$, (2) the transition function $\tau : \mathcal{X} \times \mathcal{A} \to \mathcal{X}$, (3) the reward predictor $\rho : \mathcal{X} \times \mathcal{A} \to \mathbb{R}$. Lastly, (4) a learned encoder $\phi : (\mathcal{S} \times \mathcal{A} \times \mathbb{R})^{t-1} \times \mathcal{S} \to \mathcal{Z}$, mapping a history of transitions to an environment embedding $\mathcal{Z} \in \mathbb{R}^k$: $z_i = \phi(h_{0 \to t-1})$, where $h_{0 \to t-1} = (s_0, a_0, r_1, s_1, a_1, r_2, \ldots, s_{t-1}, a_{t-1}, r_t, s_t)$. Figure 1 (right) shows a schematic overview of the model architecture.

**Policy** We denote the policy as $\pi : \mathcal{X} \to \mathcal{A}$. The *value function* $V^\pi(x)$ is defined as the expected return when starting from abstract state $x$ and following policy $\pi$:

$$V^\pi(x) = \mathbb{E}_\pi \left[ \sum_{t=0}^{\infty} \gamma^t r_t \,\middle|\, x_0 = x \right].$$

The *Q-function* $Q^\pi(x, a)$ denotes the expected return when taking action $a$ in abstract state $x$, and thereafter following $\pi$:

$$Q^\pi(x, a) = \mathbb{E}_\pi \left[ \sum_{t=0}^{\infty} \gamma^t r_t \,\middle|\, x_0 = x, a_0 = a \right].$$

**Goal** The aim is to learn a policy that maximizes the expected return across tasks sampled from the task distribution $p(\mathcal{T})$. Formally, we optimize:

$$\max_{\pi, \theta} \ \mathbb{E}_{\mathcal{T} \sim p(\mathcal{T})} \left[ \mathbb{E}_{(x_0, a_0, r_1, \ldots) \sim \pi, \mathcal{T}} \left[ \sum_{t=0}^{\infty} \gamma^t r_t \right] \right],$$

where the expectation is taken over both the task distribution and the rollout distribution induced by the policy $\pi$ across a task distribution, operating in the abstract state space $\mathcal{X}$.

## 3.1 Task-Aware Augmented State-Representations in Meta-RL

**Necessity of task-aware ARMs in meta-RL** Despite the capacity of representation models to generalize well in simple meta-RL settings such as a variety of different mazes, we argue that only through the addition of task-specific information can their potential be fully realized, a feat well-established in standard meta-RL literature Beck et al. (2025); Ni et al. (2024). Within our framework specifically, the intuition is that without task-awareness (measured by how much task information is retained by the encoder), the abstract representation model will not be able to learn a representation that can differentiate based solely on the input it receives. As such, task-agnostic state-space encoders essentially introduce a many-to-one mapping, which collapses into a single point during optimization. We provide an illustration of this insight in Figure 2.

**How can task-awareness be achieved?** We introduce a novel approach that integrates task information directly into the abstract state space $\mathcal{X}$ via the encoder $\psi$, producing representations to be *both* compact *and* task-relevant. Unlike prior meta-RL methods that condition downstream components (such as dynamics, reward models, or policies) on context (e.g., VariBAD, PEARL), our method performs *early* context injection by embedding task identity during the state abstraction phase. This design allows our agent to resolve task ambiguity upstream, providing a strong inductive bias and leading to more robust representations that generalize better to unseen tasks.

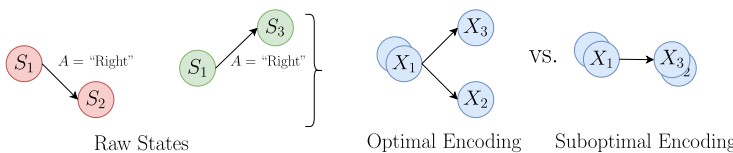

Figure 2: Illustration of the main intuition that one-to-one task-aware mappings produce better encodings. Raw input states, where $S_1$ maps to state $S_2$ in task red and to $S_3$ in task green. **Left:** Optimal encoding where both follow-up abstract states are mapped to different regions in space. **Right:** Suboptimal encoding we would expect if the encoder does not retain contextual information. In that case, we may observe a collapse of the two states, because $\psi(s_3^{\mathcal{T}_1})$ and $\psi(s_2^{\mathcal{T}_2})$ are pushed to the same point in order to minimize the loss.

By shifting task uncertainty into the encoder, we also simplify the overall architecture and training objective, eliminating the need for amortized inference or latent variable optimization at test time. Furthermore, this abstraction-first formulation is highly modular and integrates seamlessly with both model-based and model-free RL algorithms. We formally justify this design through the lens of entangled versus disentangled modeling: in the latter, state abstraction and task representation are learned separately, whereas in the former — our setting — they are integrated into a single representation. We then empirically show that our approach leads to improved generalization across standard meta-RL benchmarks.

**Proposition 1 (Entanglement Yields Lower Entropy)** *Let $\mathcal{T}_1$ and $\mathcal{T}_2$ be two tasks such that for some state-action pair $(s_t, a_t)$, the transitions differ: $P_1(s_{t+1} \mid s_t, a_t) \neq P_2(s_{t+1} \mid s_t, a_t)$. Let $z = \phi(h_{i,:t}) \in \mathcal{Z}$ be a deterministic task embedding computed from transition history. Consider two model variants: (1) a task-state encoder, where $x_t = \psi(s_t, z)$, the transition prediction is given by $x_{t+1} \approx x_t + \tau(x_t, a_t)$ and the estimated reward $\hat{r}_t = \rho(x_t, a_t)$; and (2) a state encoder, where $x_t = \psi^*(s_t)$, $x_{t+1} \approx x_t + \tau^*(x_t, a_t, z)$ and the estimated reward $\hat{r}_t = \rho^*(x_t, a_t, z)$. Then, assuming $\psi$, $\rho$ and $\tau$ as well as $\psi^*$, $\rho^*$ and $\tau^*$ are deterministic and trained to perfectly fit the transition and reward losses, the total entropy of the representations satisfies:*

$$\mathbb{H}\big(\psi(s_t, z)\big) \ \leq \ \mathbb{H}\big((\psi^*(s_t), z)\big).$$

*That is, the task-state version yields lower joint entropy due to resolving task ambiguity in the representation, resulting in simpler transitions and reward functions.*

We provide the full proof in Appendix C. Intuitively, Proposition 1 states that entanglement causes context ambiguity to be resolved *before* it reaches downstream tasks (e.g., before computing transition probabilities or state-to-action mappings via the policy). As such, following a general "conditioning reduces entropy" rule (Cover & Thomas, 1999; Gray, 2011; MacKay, 2003), which has been explored extensively in the machine learning literature (Bounoua et al., 2025; Pandey & Dukkipati, 2017; Shamir et al., 2010), early conditioning effectively ensures that all model components are one-to-one mappings. We call the task-aware abstract state space produced by the the state-task encoder the *augmented abstract* state space.

## 4 EMERALD: Architecture & Objectives

**ARM: Architecture Blocks** We introduce a parameterized abstract representation model, consisting of (1) a state-space encoder $\psi$, (2) a latent transition function $\tau$, (3) a reward predictor $\rho$, and (4) a transitions-to-context function $\phi$ to adjust for environment-specific dynamics (see Appendix G for a more detailed overview). Formally, we have four parameterized model components:

$$\psi\big(s_t, z; \theta_\psi\big) \ \rightarrow \ x_t, \quad x_{t+1} \ = \ x_t + \tau\big(x_t, a_t; \theta_\tau\big), \quad \hat{r}_t \ = \ \rho\big(x_t, a_t; \theta_\rho\big), \quad z \ = \ \phi\big(h_t; \theta_\phi\big).$$

We combine the model parameters into a jointly optimizable model via $\theta$, where $\theta = (\theta_\psi, \theta_\tau, \theta_\rho, \theta_\phi)$.

**ARM: Objectives** The transition and reward dynamics are effectively captured by the loss functions defined below. We define the transition and reward losses as:

$$\mathcal{L}(\theta)_{\text{transition}} = \mathbb{E}_{(s_t, a_t, s_{t+1}, r_t) \sim D}\left[\left\|\psi(s_{t+1}, z; \theta_\psi) - (\psi(s_t, z; \theta_\psi) + \tau(\psi(s_t, z; \theta_\psi), a_t; \theta_\tau))\right\|^2\right],$$

$$\mathcal{L}(\theta)_{\text{reward}} = \mathbb{E}_{(s_t, a_t, s_{t+1}, r_t) \sim D}\left[\left\|r_t - \rho(\psi(s_t, z; \theta_\psi), a_t; \theta_\rho)\right\|^2\right].$$

The formulations above follow standard practice (e.g., François-Lavet et al. (2019)), but we introduce context vectors $z$ retrieved from the context encoder $\phi$ and include it directly into the optimization objective. To prevent a potential state space collapse, we also include a regularizer:

$$\mathcal{L}(\theta)_{\text{reg}} = \exp\left(-C_d\left\|\psi(s^+, z; \theta_\psi) - \psi(s^-, z; \theta_\psi)\right\|^2\right),$$

where $s^+$ and $s^-$ are two randomly sampled states and $C_d$ is a constant. Intuitively, this term enforces that any two states are some distance apart.

Putting everything together, we obtain the following joint objective:

$$\mathcal{L}(\theta) = \mathcal{L}(\theta)_{\text{transition}} + \mathcal{L}(\theta)_{\text{reward}} + \beta\,\mathcal{L}(\theta)_{\text{reg}},$$

where $\beta \geq 0$ determines the regularization strength. We minimize this objective using standard mean squared error losses for the transition and reward terms. Algorithm 1 shows the pseudocode for the ARM training process. Throughout the experimental section, we sample the offline data for training (line 1 of the algorithm) the model using a random policy. However, in practice, the offline data might also be collected via a non-random or expert policy. A Python implementation can be found at `https://github.com/ljsmalbil/EMERALD`.

**Training of the policy** The policy is trained after ARM training. We freeze the representation model's parameters and pass the online transitions through the model. The augmented abstract state approach allows us to combine the learned representations with any reinforcement learning algorithm (e.g., SAC, PPO). We define the optimal policy as:

$$\pi^* = \arg\max_\pi V^\pi(\psi(s_t, z; \theta_\psi)), \quad \forall s_t \in \mathcal{S}.$$

Algorithm 2 shows the pseudo-code for policy training (full version shown in Appendix D).

## 5 Experiments

Our evaluation consists of four complementary experimental sets, each designed to test a distinct aspect of EMERALD. First, we compare EMERALD with strong baselines on *out-of-distribution* tasks following the evaluation protocol of Lee et al. (2020). Second, we compare EMERALD to another set of baselines on the more recent Meta-World ML1 and ML10 suites (Yu et al., 2020). Third, we evaluate the effect of training on multiple environments by measuring performance on various tasks. Lastly, we perform two ablation studies: (i) we examine how the quality of the ARM affects policy learning, and (ii) we visualize the learned task-aware latent space to assess how effectively the policy exploits it.

### 5.1 Comparative benchmarks

**Setting 1**: We largely follow the experimental protocol of Lee et al. (2020), using variants of the continuous-control CartPole task and several MuJoCo benchmarks (Todorov et al., 2012). The learning algorithms are trained on a set of configurations and evaluated on previously unseen ones. For CartPole, for example, we train on pole lengths 1.0, 1.5, and 2.0 and test the out-of-distribution performance on poles of length 0.5 and 2.5. HalfCheetah (Volume) is a HalfCheetah setting where we shrink or expand the Cheetah's volume, depending on the exact treatment regimen. The three evaluation domains are illustrated in Figure 3 (and

---

**Algorithm 1** EMERALD Abstract Representation Model Training

---

**Require:** ARM (transition encoder, reward predictor, abstract state encoder, task-encoder), task distribution $p(\mathcal{T})$, learning rate $\alpha_\theta$, batch size $B$, epochs per task $n_{\text{epochs}}$, initialize replay buffers $\mathcal{D}_i$ for every task $\mathcal{T}_i$, context horizon $H$

1: **Offline Data Collection**: For each task $\mathcal{T}_i$, collect a dataset $\mathcal{D}_i^{\text{offline}} = \{(s_t, a_t, s_{t+1}, r_t)\}_{t=1}^{m_i}$ of offline transitions following policy $\pi$.

2: **while** not converged **do**            ▷ ARM training loop

3:     **for** each task $\mathcal{T}_i$ **do**

4:        Reset *context window*: $h^i \leftarrow \emptyset$

5:        **for** epoch = 1 to $n_{\text{epochs}}$ **do**

6:           **for** each batch $\{(s_t, a_t, s_{t+1}, r_t)\} \sim \mathcal{D}_i^{\text{offline}}$ **do**

7:             Sample task embedding $z \sim \phi(\cdot \mid h^i; \theta_\phi)$

8:             Encode states: $x_t = \psi(s_t, z; \theta_\psi)$     $x_{t+1}^{\text{true}} = \psi(s_{t+1}, z; \theta_\psi)$

9:             Predict transition and reward: $\hat{x}_{t+1} = x_t + \tau(x_t, a_t; \theta_\tau)$

10:            Predict reward: $\hat{r}_t = \rho(x_t, a_t; \theta_\rho)$

11:            Compute loss: $\mathcal{L}_\theta = \mathcal{L}_{\text{transition}} + \mathcal{L}_{\text{reward}} + \beta \, \mathcal{L}_{\text{reg}}$

12:            Update $\psi$ parameters: $\theta_\psi \leftarrow \theta_\psi - \alpha_{\theta_\psi} \frac{1}{B} \sum_{i=1}^{B} \nabla_{\theta_\psi} \mathcal{L}_i$     ▷ Parameter updates

13:            Update $\phi$ parameters: $\theta_\phi \leftarrow \theta_\phi - \alpha_{\theta_\phi} \frac{1}{B} \sum_{i=1}^{B} \nabla_{\theta_\phi} \mathcal{L}_i$

14:            Update $\rho$ parameters: $\theta_\rho \leftarrow \theta_\rho - \alpha_{\theta_\rho} \frac{1}{B} \sum_{i=1}^{B} \nabla_{\theta_\rho} \mathcal{L}_i$

15:            Add to context window: $c^i \leftarrow c^i \cup \{(s_t, a_t, r_t, s_{t+1})\}$ until $|h^i| \leq H$

16:           **end for**

17:        **end for**

18:     **end for**

19: **end while**

20: **return** Trained ARM.

---

**Algorithm 2** EMERALD Policy Learning

---

**Require:** Trained ARM, tasks $\mathcal{T}_i \sim p(\mathcal{T})$, learning rates $\alpha_\pi, \alpha_V$, batch size $B$, rollout length $n_r$, updates $n_u$, horizon $H$; parameters $\theta_\pi, \theta_V$; buffers $\mathcal{D}_i \leftarrow \emptyset$

1: **while** not converged **do**

2:     **for** all $\mathcal{T}_i$ **do**            ▷ Rollout

3:        $\mathcal{B} \leftarrow \emptyset$, $h^i \leftarrow \emptyset$

4:        **for** $t = 1 \ldots n_r$ **do**

5:           $z \sim \phi(\cdot \mid h^i; \theta_\phi)$; $x_t = \psi(s_t, z; \theta_\psi)$, $a_t \sim \pi_{\theta_\pi}(\cdot \mid x_t)$

6:           env step $\rightarrow s_{t+1}, r_t$, $x_{t+1} = \psi(s_{t+1}, z; \theta_\psi)$

7:           add $(x_t, a_t, r_t, x_{t+1})$ to $\mathcal{B}$ and $h^i$ (truncate to $H$)

8:        **end for**

9:        $\mathcal{D}_i \leftarrow \mathcal{D}_i \cup \mathcal{B}$

10:     **end for**

11:     **for** $u = 1 \ldots n_u$ **do**            ▷ Update

12:        **for** all $\mathcal{T}_i$ **do**

13:           minibatch $b^i \sim \mathcal{D}_i$; compute $L_\pi^i, L_V^i$

14:        **end for**

15:        $\theta_\pi \leftarrow \theta_\pi - \alpha_\pi \nabla_{\theta_\pi} \sum_i L_\pi^i$; $\theta_V \leftarrow \theta_V - \alpha_V \nabla_{\theta_V} \sum_i L_V^i$

16:     **end for**

17: **end while**

18: **return** $\pi_{\theta_\pi}$, $V_{\theta_V}$

---

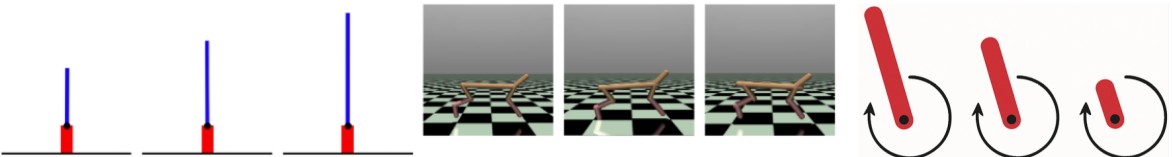

Figure 3: Illustration of the Experimental Setup. **Left:** Cartpole with different sizes. **Middle:** Three HalfCheetah with different volumes. **Right:** Pendulum with different lengths.

details are provided in Appendix L). EMERALD is paired with Soft Actor-Critic (SAC) (Haarnoja et al., 2018) (EMERALD SAC) and Proximal Policy Optimization (PPO) (Schulman et al., 2017) (EMERALD PPO). The ARM is pre-trained offline with $5 \times 10k$ CartPole transitions, $11 \times \sim 4.5k$ Pendulum transitions, and $5 \times 100k$ HalfCheetah transitions. In order not to give our model an unfair advantage, we collected the offline transitions under a random policy (ablation study in Appendix H shows the effect of using an expert policy).

Following Lee et al. (2020), we evaluate all methods on unseen environments under two regimes (see Appendix J for complete specifications):

1. **Moderate**: Unseen test tasks differ only slightly from the training distribution.

2. **Extreme**: Unseen test tasks deviate substantially from training task.

**Setting 2**: In the second setting, we compare EMERALD to the same set of baselines on the Meta-World ML10 suite Yu et al. (2020). Meta-World is a benchmark environment for evaluating multi-task and meta-reinforcement learning algorithms, consisting of a diverse collection of robotic manipulation tasks built on the MuJoCo simulator. ML10 includes ten distinct training tasks designed to assess generalization across five test task distributions.

**Setting 3**: In this setting, we compare the performance of EMERALD and VariBAD on the HalfCheetah-Direction task, a standard benchmark in meta-reinforcement learning. Unlike Settings 1 and 2, which test out-of-distribution generalization, this experiment evaluates in-distribution adaptation — specifically, how well the agent adjusts to moving in opposite directions (forward versus backward).

We compare EMERALD PPO and EMERALD SAC with ten baselines:

1. **PPO** (Schulman et al., 2017): Standard Proximal policy optimization (PPO).

2. **SAC** (Haarnoja et al., 2018): Standard Soft Actor Critic (SAC).

3. **Stacked PPO** (Lee et al., 2020): PPO variant that feeds a fixed window of past transitions to the policy.

4. **PEARL** (Rakelly et al., 2019): Infers a context variable by maximizing expected return.

5. **PPO-EP** (Zhou et al., 2019): PPO version that concatenates an embedding from early interactions to the state.

6. **CaDM** (Lee et al., 2020): Augments a forward–backward dynamics model with a context encoder.

7. **VariBAD** (Zintgraf et al., 2021): Uses an RNN to produce a belief state that is concatenated with the environment state and passed to both the policy and value functions.

8. **RMF-RL** (Ni et al., 2021): Recurrent meta-RL model-free approach with optimized architecture and hyperparameters.

9. **RNN-HN** (Beck et al., 2023): Recurrent hypernetwork approach for meta-RL.

10. **VI-HN** (Beck et al., 2023): Variational hypernetwork approach for meta-RL.

To identify the effect of the learning algorithm (PPO or SAC), we pair the most important competitors to a learning algorithm.[34] Moreover, to ensure fairness, the ARM is pretrained on offline data and the corresponding transitions are deducted from each method's overall interaction budget, which guarantees that all agents experience the same total number of transitions (see Appendix M for the complete allocation budget). We further use the same hyperparmeter tuning process for all baselines (Appendix N).

(a) PPO-based methods

| | CartPole (Lengths) | | Pendulum (Lengths) | | HalfCheetah (Volume) | |
|---|---|---|---|---|---|---|
| | Moderate | Extreme | Moderate | Extreme | Moderate | Extreme |
| PPO | **198.2** $\pm$ 0.9 | 187.8 $\pm$ 4.7 | -1113.2 $\pm$ 69.1 | -1356.8 $\pm$ 48.0 | 807.7 $\pm$ 553.6 | 574.0 $\pm$ 645.6 |
| Stacked PPO | 197.8 $\pm$ 1.3 | 189.2 $\pm$ 6.1 | -475.7 $\pm$ 228.1 | -488.2 $\pm$ 178.2 | 361.1 $\pm$ 141.7 | 5.7 $\pm$ 208.1 |
| PPO EP | 196.3 $\pm$ 4.0 | 184.5 $\pm$ 9.7 | -374.3 $\pm$ 24.6 | **-256.7** $\pm$ 26.4 | 895.3 $\pm$ 445.1 | 674.2 $\pm$ 686.8 |
| CaDM (PPO) | 197.9 $\pm$ 3.0 | 193.0 $\pm$ 3.5 | **-279.8** $\pm$ 42.1 | -426.4 $\pm$ 227.0 | 1224.2 $\pm$ 630.0 | 1021.1 $\pm$ 676.6 |
| VariBAD (PPO) | **199.1** $\pm$ 2.8 | 192.0 $\pm$ 3.2 | **-318.5** $\pm$ 56.6 | -643.1 $\pm$ 311.4 | **2964.0** $\pm$ 179.5 | 1618.3 $\pm$ 251.8 |
| RMF-RL (PPO) | 198.3 $\pm$ 0.9 | 191.9 $\pm$ 2.4 | -505.6 $\pm$ 42.5 | -899.3 $\pm$ 185.3 | 2303.5 $\pm$ 146.3 | 1037.0 $\pm$ 268.1 |
| VI-HN (PPO) | 197.7 $\pm$ 2.7 | 194.5 $\pm$ 5.3 | -421.1 $\pm$ 56.8 | -531.3 $\pm$ 121.3 | 1882.1 $\pm$ 91.5 | 1167.9 $\pm$ 185.2 |
| RNN-HN (PPO) | 198.7 $\pm$ 2.1 | 193.7 $\pm$ 4.1 | -444.9 $\pm$ 51.0 | -497.8 $\pm$ 117.2 | 1982.9 $\pm$ 88.2 | 984.2 $\pm$ 242.5 |
| EMERALD (PPO) | **199.1** $\pm$ 1.1 | **199.3** $\pm$ 3.5 | **-312.4** $\pm$ 19.0 | -313.7 $\pm$ 101.5 | **2942.5** $\pm$ 213.5 | **1401.4** $\pm$ 421.3 |

(b) SAC-based methods

| | CartPole (Lengths) | | Pendulum (Lengths) | | HalfCheetah (Volume) | |
|---|---|---|---|---|---|---|
| | Moderate | Extreme | Moderate | Extreme | Moderate | Extreme |
| SAC | **199.2** $\pm$ 0.3 | **199.4** $\pm$ 0.1 | -402.0 $\pm$ 131.3 | -1274.2 $\pm$ 133.7 | 1998.3 $\pm$ 142.1 | 177.4 $\pm$ 321.3 |
| PEARL (SAC) | 198.0 $\pm$ 1.4 | 187.5 $\pm$ 10.9 | -645.3 $\pm$ 320.7 | -1136.6 $\pm$ 251.0 | 642.1 $\pm$ 488.3 | 462.1 $\pm$ 534.5 |
| VariBAD (SAC) | 194.7 $\pm$ 4.5 | 189.6 $\pm$ 1.7 | -1001.3 $\pm$ 75.1 | -1193.8 $\pm$ 94.1 | 1755.6 $\pm$ 194.2 | 1424.9 $\pm$ 258.3 |
| RMF-RL (SAC) | 198.4 $\pm$ 2.1 | 197.2 $\pm$ 3.1 | -769.3 $\pm$ 198.1 | -1058.5 $\pm$ 238.1 | 4419.1 $\pm$ 219.3 | 2142.2 $\pm$ 323.6 |
| VI-HN (SAC) | 199.2 $\pm$ 3.6 | 196.2 $\pm$ 6.4 | -657.4 $\pm$ 128.4 | -1098.6 $\pm$ 202.3 | 4365.4 $\pm$ 307.9 | **2744.3** $\pm$ 439.4 |
| RNN-HN (SAC) | 196.3 $\pm$ 2.3 | 194.7 $\pm$ 3.0 | -738.4 $\pm$ 271.9 | -991.3 $\pm$ 298.1 | 2837.7 $\pm$ 218.0 | 2539.3 $\pm$ 260.8 |
| EMERALD (SAC) | 194.6 $\pm$ 2.7 | **196.2** $\pm$ 4.9 | **-198.3** $\pm$ 23.3 | **-749.4** $\pm$ 340.9 | **5101.1** $\pm$ 151.8 | **3271.8** $\pm$ 180.2 |

Table 1: Performance comparison means over 8 seeds (Mean Return $\pm$ Std.) for CartPole, Pendulum, and HalfCheetah (Volume). The best statistically significant (Welch's t-test $p < 0.01$) results in each group are boldfaced and quantify the rolling average over the last 100 timesteps.

## 5.2 Results Benchmark Experiments

**Results for Setting 1**: Table 1b shows that EMERALD (SAC) surpasses most SAC-based baselines; the corresponding learning curves are displayed in Figures 4a–4b. In the PPO-based setting, we achieve performance on par with VariBAD (PPO). On the Pendulum task, EMERALD (PPO) does not outperform PPO-EP in the extreme regime, suggesting a potential limitation when the environment exhibits strong periodic dynamics. However, the difference between EMERALD (PPO) and PPO-EP is not large. Notably, both

---

[3]N.B. For the hypernetwork-based approaches (VI-HN and RNN-HN) as well as VariBAD, we could not find any official SAC implementations. In prior work these methods are typically paired with on-policy algorithms such as PPO, and empirical results with off-policy algorithms like SAC are rarely reported. For completeness of comparison, we implemented SAC variants of these baselines ourselves using the existing PPO-based source code.

[4]Appendix O contains an analysis whereby we treat the learning algorithm as a hyperparameter.

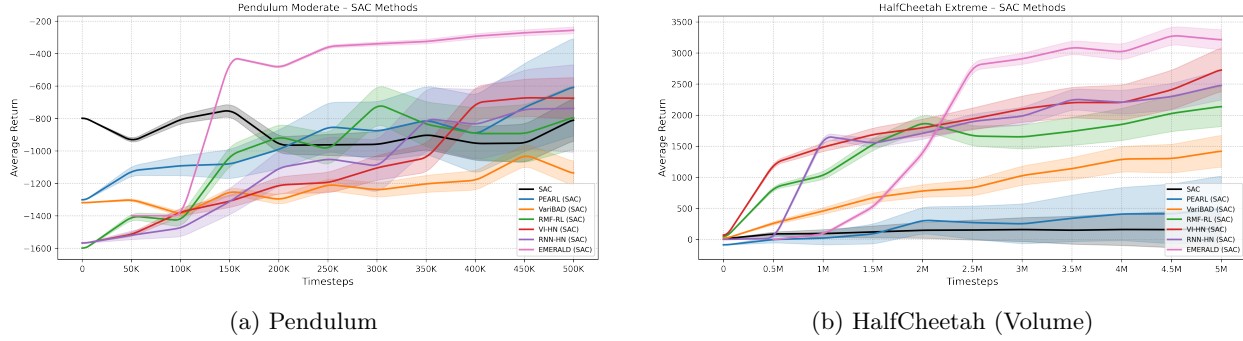

(a) Pendulum                   (b) HalfCheetah (Volume)

Figure 4: Learning Curves for two out-of-distribution Experiments from the Moderate Regime (rolling averages over 8 independent runs and adjusted for ARM budget allocation)

EMERALD (SAC) and EMERALD (PPO) perform well overall, underscoring the benefit of modularity: one can boost performance by combining the ARM with a different learning algorithm. Interestingly, the performance of the non-meta RL learners appears to be predictive of performance with EMERALD: for instance, SAC and EMERALD (SAC) both outperform other methods on the HalfCheetah tasks. Furthermore, while VariBAD does achieve higher performance when paired with PPO, in the SAC-based setting EMERALD (SAC) achieves much higher performance, suggesting that our approach is more naturally paired with SAC than VariBAD.

| (a) PPO-based methods | | (b) SAC-based methods | |
|---|---|---|---|
| **Method** | **HalfCheetah (Direction)** | **Method** | **HalfCheetah (Direction)** |
| PPO | -219.7 ± 75.2 | SAC | -173.9 ± 58.2 |
| VariBAD (PPO) | **2112.5** ± 193.6 | VariBAD (SAC) | 2806.3 ± 118.7 |
| EMERALD (PPO) | **2259.9** ± 317.1 | EMERALD (SAC) | **3343.2** ± 291.1 |

Table 2: Performance over 8 seeds (Mean Return ± Std.) with random seeds.

**Results for Setting 2**: Across both Meta-World ML10 and ML1, EMERALD variants generally achieve strong and stable performance compared to existing meta-reinforcement learning baselines. On ML10, EMERALD (SAC) attains the highest average performance in both training and testing settings, indicating effective adaptation across multiple tasks. EMERALD (PPO) performs slightly below its SAC counterpart but still matches or exceeds other baselines such as PEARL and RMF-RL on most metrics. Overall, these results suggest that EMERALD's framework contributes to improved consistency and generalization, with SAC providing a performance edge in both efficiency and asymptotic returns. Interestingly, when considering the other baselines, we observe that performance drops significantly when using SAC. For example, VI-HN (PPO) reaches a training score of 41.8%, whereas its SAC counterpart achieves barely half of that. We observe a similar trend on the SAC-based versions on ML1, where for the hypernetworks (RNN-HN (SAC) and VI-HN (SAC)) as well as VariBAD, underperform compared to their PPO-based counterparts. We believe this may be due to the fact that architectures rely on an on-policy, trajectory-consistent data distribution, while SAC's off-policy replay buffer breaks this temporal coherence and destabilizes both latent adaptation and hypernetwork weight generation (see Appendix N for a more extensive discussion). EMERALD, on the other hand, shows a relatively stable performance, regardless of the chosen policy.

**Results for Setting 3:** The results are reported in Table 2 and show that the PPO version of our approach performs on par with VariBAD (PPO). The improvement over regular PPO further suggests that training *with* augmented abstract states provides an additional boost in performance in this experimental setting. Similar to setting 1, we observe that EMERALD SAC outperforms VariBAD (SAC).

(a) PPO-based methods

| Method | Avg (Train) | Avg (Test) |
|---|---|---|
| RMF-RL (PPO) | $16.7 \pm 1.0$ | $13.8 \pm 0.4$ |
| VariBAD (PPO) | $\mathbf{52.4} \pm 2.9$ | $\mathbf{16.1} \pm 1.2$ |
| VI-HN (PPO) | $44.3 \pm 1.9$ | $14.2 \pm 1.1$ |
| RNN-HN (PPO) | $14.7 \pm 0.8$ | $12.3 \pm 0.3$ |
| EMERALD (PPO) | $46.7 \pm 1.9$ | $15.3 \pm 0.9$ |

(b) SAC-based methods

| Method | Avg (Train) | Avg (Test) |
|---|---|---|
| RMF-RL (SAC) | $48.5 \pm 1.4$ | $10.7 \pm 0.9$ |
| VariBAD (SAC) | $16.6 \pm 0.7$ | $13.1 \pm 1.1$ |
| VI-HN (SAC) | $20.8 \pm 0.4$ | $14.0 \pm 0.6$ |
| RNN-HN (SAC) | $0.0 \pm 0.0$ | $0.0 \pm 0.0$ |
| EMERALD (SAC) | $\mathbf{53.9} \pm 0.6$ | $\mathbf{17.6} \pm 1.0$ |

Table 3: Performance comparison on Meta-World ML10. Results are reported as mean percentages $\pm$ standard error over 8 seeds.

(a) PPO-based methods

| Method | door-open | basketball | window-open | pick-place | button-topdown |
|---|---|---|---|---|---|
| RMF-RL (PPO) | $0.0 \pm 0.0$ | $0.0 \pm 0.0$ | $34 \pm 1.0$ | $0.0 \pm 0.0$ | $0.0 \pm 0.0$ |
| VariBAD (PPO) | $\mathbf{91} \pm 2.4$ | $0.0 \pm 0.0$ | $\mathbf{100} \pm 0.0$ | $28 \pm 2.0$ | $98 \pm 1.0$ |
| VI-HN (PPO) | $0.0 \pm 0.0$ | $0.0 \pm 0.0$ | $\mathbf{100} \pm 0.0$ | $\mathbf{33} \pm 0.3$ | $0.0 \pm 0.0$ |
| RNN-HN (PPO) | $0.0 \pm 0.0$ | $0.0 \pm 0.0$ | $95 \pm 3.5$ | $0.0 \pm 0.0$ | $0.0 \pm 0.0$ |
| EMERALD (PPO) | $\mathbf{92} \pm 0.6$ | $\mathbf{4.0} \pm 0.9$ | $65 \pm 3.8$ | $0.0 \pm 0.0$ | $\mathbf{99} \pm 1.0$ |

(b) SAC-based methods

| Method | door-open | basketball | window-open | pick-place | button-topdown |
|---|---|---|---|---|---|
| RMF-RL (SAC) | $99 \pm 0.3$ | $\mathbf{40} \pm 0.6$ | $\mathbf{96} \pm 0.6$ | $0.0 \pm 0.0$ | $98 \pm 0.7$ |
| VariBAD (SAC) | $0.0 \pm 0.0$ | $0.0 \pm 0.0$ | $0.0 \pm 0.0$ | $0.0 \pm 0.0$ | $0.0 \pm 0.0$ |
| VI-HN (SAC) | $0.0 \pm 0.0$ | $0.0 \pm 0.0$ | $0.0 \pm 0.0$ | $0.0 \pm 0.0$ | $0.0 \pm 0.0$ |
| RNN-HN (SAC) | $0.0 \pm 0.0$ | $0.0 \pm 0.0$ | $0.0 \pm 0.0$ | $0.0 \pm 0.0$ | $0.0 \pm 0.0$ |
| EMERALD (SAC) | $\mathbf{100} \pm 0.0$ | $2.0 \pm 0.5$ | $64 \pm 0.4$ | $0.0 \pm 0.0$ | $\mathbf{99} \pm 0.6$ |

Table 4: Performance comparison on Meta-World ML1 tasks. Results are reported as mean percentages $\pm$ standard error over 8 seeds

## 5.3 Ablation Studies

**Effect of Environment Diversity on Performance** We ask whether training on a *greater diversity* of tasks is more beneficial than training with *more data* drawn from a single environment. This is an in-distribution setting, as train and test tasks are the same. We allocate a total budget of 3M transitions for HalfCheetah (Volume) and 200k for CartPole. In **Configuration 1** the entire budget is spent on one task, whereas in **Configuration 2** the budget is split evenly across multiple tasks. Table 5 summarizes the outcomes. Configuration 2 yields higher overall performance, indicating that exposure to a wider set of environments outweighs simply scaling up data for a single task, demonstrating that increased task-diversity offers greater sample efficiency.

**Model Effect on Policy Performance** We study how performance varies with (i) the number of training epochs, (ii) the amount of offline data, and (iii) the presence of task-aware encodings. Figure 5 reports the results (each marker is the mean of 8 seeds). In the CartPole setting with 10k transitions, we observe a slightly higher performance of the task-aware variant. Similarly, for CartPole with 100k transitions, the task-aware variant outperforms the task-agnostic one. In the HalfCheetah (Volume) setting, the same pattern emerges but is more pronounced: both the task-aware 100k and 10k-transition regimes converge earlier. Moreover, the performance of the task-agnostic (10k) variant is effectively zero, which suggests that removing task encodings (i.e., using task-agnostic abstract states) sharply degrades performance, particularly on HalfCheetah, demonstrating the effectiveness of using a state-task encoder.

| Environment | Config | # Training Tasks | Samples per Task | Mean Return ± Std |
|---|---|---|---|---|
| CartPole | 1 | 1 | 200k | $91.4 \pm 37.5$ |
| CartPole | 2 | 5 | 40k | $187.0 \pm 19.8$ |
| HalfCheetah | 1 | 1 | 3M | $749.2 \pm 209.3$ |
| HalfCheetah | 2 | 5 | 600k | $1218.0 \pm 338.7$ |

Table 5: EMERALD In-Distribution Performance on Training Tasks (Mean Return ± Std. over 5 seeds). Mean difference statistically significant (Welch's t-test $p < 0.01$)

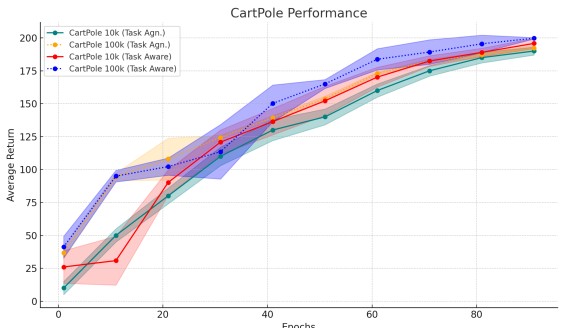
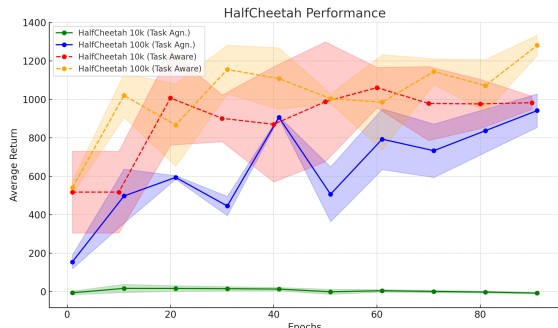

Figure 5: Average Policy Performance (8 seeds) with stdev. on CartPole and HalfCheetah (Volume) for several Training Iterations, starting from a model trained with 1 epoch up to 100 epochs. Dashed lines are task-aware iterations, solid lines display task-agnostic versions (no context encoding).

**Latent Trajectory Analysis** Figure 6 visualizes policy behavior in both training and test environments. For each environment, we record the first 50 transitions and project them into a three-dimensional principal component space. The resulting training and test trajectories exhibit similar shapes but occupy distinct regions, suggesting that EMERALD learns behaviors that transfer across tasks while remaining sensitive to task identity.

# 6 Discussion & Conclusion

We introduced EMERALD, a meta-reinforcement-learning method that leverages augmented, context-aware abstract states. Across benchmarks, our approach either outperforms other model-free context-based baselines or performs on par with state-of-the art approaches. In particular, we identified the combination of EMERALD with SAC as a natural pairing, performing generally better than competitors. On the other hand, when using PPO the performance of EMERALD compared to other PPO-based baselines is less pronounced. Still, the fact that EMERALD easily pairs with both on-policy and off-policy methods, shows its versatility. Further, relative to its non-meta-RL counterparts, we show that task-aware representation models yield substantial gains.

Beyond improved performance, we provided justification for the representational and architectural advantages of our approach: by disentangling task-relevant features early in the pipeline, we simplify training, improve robustness, and gain compatibility with a broad class of RL algorithms as our procedure yields a simple encoder that produces task-aware abstractions. Despite these strengths, EMERALD has limitations. First, the optimal structure of the latent space (e.g., dimensionality) can be environment-dependent and difficult to determine *a priori*. Second, as the number of training tasks grows, the latent space may need to expand to capture task diversity, potentially increasing computational cost. In such cases, hybrid approaches that decouple task inference from state abstraction may offer better scalability. Identifying where this trade-off becomes critical is an important direction for future work.

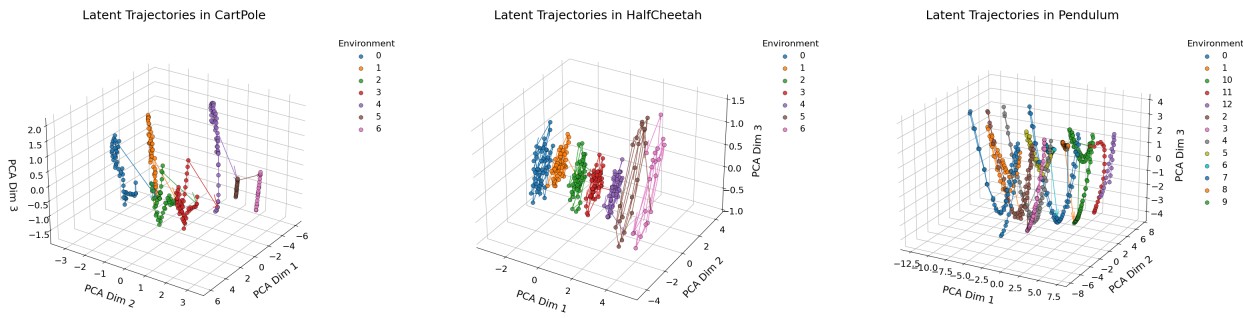

Figure 6: Visualization of the Latent Space (PCA mapping) of the EMERALD ARM with Training and Test Environments of the first 50 transitions. **Left:** Cartpole. **Middle:** HalfCheetah (Volume). **Right:** Pendulum. In CartPole and HalfCheetah (Volume) the out-of-distribution trajectories are 5 and 6, while in Pendulum 11 and 12.

A natural extension of EMERALD is to integrate it into a *model-based* framework. Whereas the present study focuses on using a *pretrained* latent model within a modular, model-free pipeline, one could instead exploit the latent dynamics directly to train a policy, thereby opening the door to direct comparisons with established model-based meta-RL methods.

## Acknowledgments

We thank the reviewers and the editor for their substantial time, valuable and extensive discussions, and the care they dedicated to improving this work. This research was supported by the EU Horizon 2020 project grant No. 96534 (ICARE4OLD).

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

# A  Notations

- $\mathcal{S}$: State space.

- $\mathcal{A}$: Action space.

- $\mathcal{T}_i$: The $i$-th task, defined as an MDP $(\mathcal{S}, \mathcal{A}, P_i, r_i, \gamma)$.

- $P_i(s_{t+1} \mid s_t, a_t)$: Transition kernel for task $\mathcal{T}_i$.

- $r_i(s, a)$: Reward function for task $\mathcal{T}_i$.

- $\gamma \in [0, 1)$: Discount factor.

- $p(\mathcal{T})$: Distribution over possible tasks.

- $p(\mathcal{T}_{\text{train}}), p(\mathcal{T}_{\text{test}})$: Training and test distributions (subsets of $p(\mathcal{T})$).

- $\mathcal{T}_i \sim p(\mathcal{T})$: Sampling the $i$-th task from the task distribution.

- $\mathcal{D}_i = \{(s_{i,k}, a_{i,k}, s_{i,k+1}, r_{i,k})\}_{k=1}^{m_i}$: Dataset of transitions for task $\mathcal{T}_i$, with $m_i$ transitions.

- $\mathcal{D} = \bigcup_{i=1}^{N} \mathcal{D}_i$: Combined dataset across $N$ tasks.

- $\theta$: Parameters of the abstract representation model or policy, depending on context.

- $\mathcal{X}$: Abstract (latent) state space.

- $P_i(x_{t+1} \mid x_t, a_t)$: Transition kernel in the latent space for task $\mathcal{T}_i$.

- $\pi : \mathcal{X} \to \mathcal{A}$: A policy mapping from abstract states to actions.

- $\pi_\theta$: A parameterized policy with parameters $\theta$.

- $\psi(s_t, z)$: State encoder mapping raw state $s_t$ and context $z$ to abstract state $x_t \in \mathcal{X}$.

- $\phi(h_{i,:t})$: Context encoder that produces a task embedding $z \in \mathcal{Z}$ from a trajectory history.

- $\mathcal{Z}$: Context (task embedding) space.

- $\mathcal{L}(f\theta); \mathcal{T}_i)$: Loss of the ARM on task $\mathcal{T}_i$.

- $\mathcal{L}(\theta; \mathcal{T})$: Loss of the ARM over a set of tasks $\mathcal{T}$. If clear from context, we write $\mathcal{L}(\theta)$ to denote the expected loss over $p(\mathcal{T})$.

- $\hat{\mathcal{L}}(\theta; \mathcal{T})$: Empirical loss of the ARM over a set of tasks $\mathcal{T}$.

- $\mathcal{F}$: Function class.

- $\Theta$: Parameter space.

- $\mathbb{H}(X)$: Entropy of a random variable $X$.

# B   Definitions

**Definition 1** *We define a task as a Markov Decision Process (MDP) via the following 5-tuple*

$$\mathcal{T}_i = (\mathcal{S}, \mathcal{A}, P_i, r_i, \gamma),$$

*where:*

- $\mathcal{S}$ *denotes the state space,*

- $\mathcal{A}$ *denotes the action space,*

- $P_i(s_{t+1} \mid s_t, a_t)$ *is the transition kernel for the task* $\mathcal{T}_i \sim p(\mathcal{T})$,

- $r_i(s, a)$ *is the reward function, i.e.* $r_i : \mathcal{S} \times \mathcal{A} \to \mathbb{R}$,

- $\gamma \in [0, 1]$ *is the discount factor.*

**Definition 2** *We define four parameterized model components:*

$$\psi(s_t, z; \theta_\psi) \to x_t, \quad x_{t+1} = x_t + \tau(x_t, a_t; \theta_\tau), \quad \hat{r}_t = \rho(x_t, a_t; \theta_\rho), \quad z = \phi(h_t; \theta_\phi),$$

*where: -* $\psi : \mathcal{S} \times \mathcal{Z} \to \mathcal{X}$ *is a contextual encoder mapping high-dimensional states* $s \in \mathcal{S} \subseteq \mathbb{R}^n$ *and task embeddings* $z \in \mathcal{Z} \subseteq \mathbb{R}^k$ *to abstract states* $x \in \mathcal{X} \subseteq \mathbb{R}^d$, $\tau : \mathcal{X} \times \mathcal{A} \to \mathcal{X}$ *is a transition model predicting latent state deltas, -* $\rho : \mathcal{X} \times \mathcal{A} \to \mathbb{R}$ *predicts rewards in abstract space, and -* $\phi : (\mathcal{S} \times \mathcal{A} \times \mathbb{R})^{t-1} \times \mathcal{S} \to \mathcal{Z}$ *is a context encoder mapping past transitions to a task embedding, where parameters* $\theta = (\theta_\psi, \theta_\tau, \theta_\rho, \theta_\phi)$.

We draw $N$ i.i.d. *training tasks* $\mathcal{T}_i \sim p(\mathcal{T})$. Each task $\mathcal{T}_i$ provides a dataset $\left\{(s_{i,t}, a_{i,t}, s_{i,t+1}, r_{i,t})\right\}_{t=1}^{m_i}$. Denote the combined data by

$$\mathcal{D} = \left\{ (\mathcal{T}_i, s_{i,t}, a_{i,t}, s_{i,t+1}, r_{i,t}) \,\middle|\, i = 1, \ldots, N;\ t = 1, \ldots, m_i \right\}.$$

**Definition 3** *For each task* $\mathcal{T}_i$, *define the transition and reward losses:*

$$\mathcal{L}_{transition}(\theta; \mathcal{T}_i) = \mathbb{E}_{(s_t, a_t, s_{t+1}, r_t) \sim \mathcal{D}_i} \left[ \left\| \psi(s_{t+1}; \theta_\psi) - \left( \psi(s_t; \theta_\psi) + \tau(\psi(s_t; \theta_\psi), a_t; \theta_\tau) \right) \right\|^2 \right],$$

$$\mathcal{L}_{reward}(\theta; \mathcal{T}_i) = \mathbb{E}_{(s_t, a_t, s_{t+1}, r_t) \sim \mathcal{D}_i} \left[ \left\| r_t - \rho(\psi(s_t; \theta_\psi), a_t; \theta_\rho) \right\|^2 \right].$$

*For a scalar* $\lambda > 0$, *the total task-specific loss is*

$$\mathcal{L}(\theta; \mathcal{T}_i) = \mathcal{L}_{transition}(\theta; \mathcal{T}_i) + \lambda \mathcal{L}_{reward}(\theta; \mathcal{T}_i).$$

*We collect the parameters* $\theta = (\theta_\psi, \theta_\tau, \theta_\rho$.

*To simplify notation, we denote the expected loss over all tasks as* $\mathcal{L}(\theta)$ *rather than* $\mathcal{L}((\theta; \mathcal{T})$.

**Definition 4** *Given a Markov Decision Process* $\mathcal{T}_i = (\mathcal{S}, \mathcal{A}, P_i, r_i, \gamma)$, *a policy is a mapping* $\pi^{\mathcal{T}_i} : \mathcal{S} \times \mathcal{A} \to [0, 1]$. *We have* $\pi^{\mathcal{T}_i}(a_t \mid s_t)$, *which defines the probability of selecting action* $a_t$ *when in state* $s_t$. *A deterministic policy is a special case where the mapping reduces to* $\pi^{\mathcal{T}_i} : \mathcal{S} \to \mathcal{A}$, *meaning that for each state* $s_t$, *there exists a single action* $a = \pi^{\mathcal{T}_i}(s_t)$. *A stochastic policy assigns a probability distribution over actions, meaning that for any state* $s_t$, *the probabilities over actions sum to* 1.

**Assumption 1 (Existence of Distribution)** *We assume there is a distribution* $\mathcal{T}$ *over possible tasks indexed by* $\mathcal{T}_i$. *Each task* $\mathcal{T}_i$ *defines a stochastic mapping from* $(s_t, a_t)$ *to* $(s_{t+1}, r_t)$.

**Definition 5** *Given a Markov Decision Process (MDP) $\mathcal{T}_i = (\mathcal{S}, \mathcal{A}, P_i, r_i, \gamma)$ and a policy $\pi$, we define the value function of policy $\pi$ as a mapping $V_\pi^{\mathcal{T}_i} : \mathcal{S} \to \mathbb{R}$ such that:*

$$V_\pi^{\mathcal{T}_i}(s_t) = \mathbb{E}_\pi \left[ \sum_{t=0}^\infty \gamma^t r_i(s_t, a_t) \mid s_0 = s \right].$$

*Here, the expectation is taken over the trajectory induced by following $\pi(a_t \mid s_t)$, where the state transitions follow the transition dynamics $P_i$.*

**Definition 6** *Given a Markov Decision Process (MDP) $\mathcal{T}_i = (\mathcal{S}, \mathcal{A}, P_i, r_i, \gamma)$ and a policy $\pi$, we define the state-action value function (Q-function) of policy $\pi$ as a mapping $Q^{\pi,\mathcal{T}_i} : \mathcal{S} \times \mathcal{A} \to \mathbb{R}$ such that:*

$$Q^{\pi,\mathcal{T}_i}(s_t, a_t) = \mathbb{E}_\pi \left[ \sum_{t=0}^\infty \gamma^t r_i(s_t, a_t) \mid s_0 = s, a_0 = a \right].$$

*Here, the expectation is taken over the trajectory where the initial action is $a$ in state $s$, and subsequent actions are selected according to $\pi(a_t \mid s_t)$, while state transitions follow $P_i$.*

## C   Propositions and Lemmas with Proofs

**Proposition 1 (Entanglement Yields Lower Entropy)** *Let $\mathcal{T}_1$ and $\mathcal{T}_2$ be two tasks such that for some state-action pair $(s_t, a_t)$, the transitions differ: $P_1(s_{t+1} \mid s_t, a_t) \neq P_2(s_{t+1} \mid s_t, a_t)$. Let $z = \phi(h_{i,:t}) \in \mathcal{Z}$ be a deterministic task embedding computed from transition history. Consider two model variants: (1) a task-state encoder, where $x_t = \psi(s_t, z)$, the transition prediction is given by $x_{t+1} \approx x_t + \tau(x_t, a_t)$ and the estimated reward $\hat{r}_t = \rho(x_t, a_t)$; and (2) a state encoder, where $x_t = \psi^*(s_t)$, $x_{t+1} \approx x_t + \tau^*(x_t, a_t, z)$ and the estimated reward $\hat{r}_t = \rho^*(x_t, a_t, z)$. Then, assuming $\psi$, $\rho$ and $\tau$ as well as $\psi^*$, $\rho^*$ and $\tau^*$ are deterministic and trained to perfectly fit the transition and reward losses, the total entropy of the representations satisfies:*

$$\mathbb{H}\big(\psi(s_t, z)\big) \;\leq\; \mathbb{H}\big((\psi^*(s_t), z)\big).$$

*That is, the task-state version yields lower joint entropy due to resolving task ambiguity in the representation, resulting in simpler transitions and reward functions.*

**Proof 2** *Let $s$ and $z$ be random variables representing the state and task, respectively. Let $\psi : \mathcal{S} \times \mathcal{Z} \to \mathcal{X}$ be a* task-aware *encoder, and $\psi^* : \mathcal{S} \to \mathcal{X}$ a* task-agnostic *encoder.*

*Assume there exists a deterministic function $g$ such that*

$$\psi(s, z) = g(\psi^*(s), z),$$

*that is, the task-aware encoding can be obtained by transforming the joint variable $(\psi^*(s), z)$. We refer to the left-hand side as the early-context encoding (direct mapping to the abstract state-task space) and to the right-hand side as the late-context encoding (post-hoc adjustment within abstract state space).*

*Since $g$ is a deterministic function applied to $(\psi^*(s), z)$, the chain rule for entropy gives:*

$$\mathbb{H}(\psi^*(s), z) \;=\; \mathbb{H}\big(g(\psi^*(s), z)\big) + \mathbb{H}\big((\psi^*(s), z) \,\big|\, g(\psi^*(s), z)\big).$$

*Because conditional entropy is nonnegative, we have*

$$\mathbb{H}\big(g(\psi^*(s), z)\big) \leq \mathbb{H}(\psi^*(s), z).$$

*Substituting $\psi(s, z) = g(\psi^*(s), z)$, we obtain*

$$\mathbb{H}(\psi(s, z)) \leq \mathbb{H}(\psi^*(s), z).$$

*Equality holds if and only if $g$ is injective (i.e., information-preserving) on the support of $(\psi^*(s), z)$. Otherwise, any many-to-one mapping $g$ strictly reduces entropy.*

*Hence, the early (task-state) representation cannot have higher entropy than the late (state + task) representation; it resolves task ambiguity directly in the encoded space, yielding lower representational complexity and simpler transition/reward dynamics.* $\square$

**Remark 1 ( Conditions for strict inequality)** *In practice, our architecture will often have a strictly lower entropy than downstream task-aliasing, because it operates directly in the state–task space, avoiding the need for downstream readjustment. To illustrate this, consider the following.*

*We have*

$$\mathbb{H}(\psi(s, z)) \leq \mathbb{H}(\psi^*(s), z).$$

*Equality holds if and only if $\psi(s, z)$ and $(\psi^*(s), z)$ are one-to-one, that is, if $\psi(s, z)$ can be perfectly recovered from $(\psi^*(s), z)$ and vice versa.*

*As an example, consider two tasks $\mathcal{T}_1$ and $\mathcal{T}_2$ that differ in their transition dynamics from the same state–action pair $(s_1, a = Right)$:*

$$s_1 \xrightarrow{\mathcal{T}_1} s_2, \qquad s_1 \xrightarrow{\mathcal{T}_2} s_3.$$

*If the encoder $\psi^*$ collapses the two follow-up states into the same abstract state, equality can still hold provided that the task embedding $z$ can perfectly disambiguate this collapse.*

*However, we get a strict inequality once $\psi^*$ induces a many-to-one mapping (state aliasing) and $z$ cannot fully restore the lost information (e.g., due to overfitting on the training environments). In this case, the early task-aware representation $\psi(s, z)$ carries strictly less entropy, reflecting a more compact and disambiguated encoding of the task-conditioned dynamics.*

**Remark 2 (On Collapse and the Role of the Assumption and Regularizer)** *The entropy inequality follows from subadditivity. However, it can be trivially satisfied if $\psi$ collapses. However, the combination of the next-state encoder and the reward predictor will prevent this from happening. We additionally ensure that the representation encodes nontrivial information about the state.*

*Practically, we enforce this via a regularizer:*

$$\mathcal{L}(\theta)_{reg} = \exp\left(-C_d \left\| \psi(s^+, z; \theta_\psi) - \psi(s^-, z; \theta_\psi) \right\|^2\right),$$

*where $s^+$ and $s^-$ are randomly sampled states. This encourages separation in latent space and mitigates representational collapse.*

# D   Appendix D: Policy Algorithm

---

**Algorithm 3** EMERALD Policy Learning

---

**Require:** Trained ARM $f_\theta$, tasks $\mathcal{T}_i \sim p(\mathcal{T})$, learning rates $\alpha_\pi, \alpha_V$, batch size $B$, number of gradient updates $n_{\text{updates}}$, number of environment steps per rollout $n_{\text{rollout}}$, policy parameters $\theta_\pi$ and value/critic parameters $\theta_V$, per-task replay buffers $\mathcal{D}_i \leftarrow \emptyset$, context horizon $H$

---

1:  **while** not converged **do**                                                    ▷ Policy learning loop
2:      **for** each task $\mathcal{T}_i$ **do**                                       ▷ Data collection
3:          Reset temporary buffer: $\mathcal{B} \leftarrow \emptyset$
4:          Reset *context window*: $c^i \leftarrow \emptyset$
5:          **for** step $= 1$ to $n_{\text{rollout}}$ **do**
6:              Get $c^i$ from recent transitions; sample $z \sim \phi(z \mid c^i; \theta_\phi)$
7:              Observe state $s_t$; encode $x_t = \psi(s_t, z; \theta_\psi)$
8:              Sample action $a_t \sim \pi_{\theta_\pi}(\cdot \mid x_t)$
9:              Execute $a_t$ in $\mathcal{T}_i$; observe $(s_{t+1}, r_t)$
10:             Encode next state $x_{t+1}^{\text{true}} = \psi(s_{t+1}, z; \theta_\psi)$
11:             Store $(x_t, a_t, r_t, x_{t+1}^{\text{true}})$ in $\mathcal{B}$
12:             Update context: $c^i \leftarrow c^i \cup \{(s_t, a_t, r_t, s_{t+1})\}$ until $|c^i| \leq H$
13:         **end for**
14:         Update replay buffer: $\mathcal{D}_i \leftarrow \mathcal{D}_i \cup \mathcal{B}$
15:     **end for**

16:     **for** update $= 1$ to $n_{\text{updates}}$ **do**                            ▷ Optimization step
17:         **for** each task $\mathcal{T}_i$ **do**
18:             Sample minibatch $b^i = \{(s_t, a_t, r_t, s_{t+1})\}_{j=1}^{B} \sim \mathcal{D}_i$
19:             Compute *task-specific* policy loss $L_{\text{policy}}^i$                ▷ e.g. PPO, SAC, REINFORCE.
20:             Compute optional value loss $L_{\pi\theta_{\text{value}}}^i$
21:         **end for** $\theta_\pi \leftarrow \theta_\pi - \alpha_\pi \nabla_{\theta_\pi} \sum_i L_{\text{policy}}^i$
22:         Update value network: $\theta_V \leftarrow \theta_V - \alpha_V \nabla_{\theta_V} \sum_i L_{\text{value}}^i$
23:     **end for**
24: **end while**
25: **return** Trained policy $\pi_{\theta_\pi}$ and $V_{\theta_V}$

---

# E  Policy Algorithm Details

The goal of reinforcement learning is to learn a policy that - on the basis of the current state - takes the action that maximizes the cumulative reward. In our setting, moreover, the "raw" input states are mapped to the learned latent space. Formally, the optimal policy satisfies:

$$\pi^* = \arg\max_{\pi} V_{\pi}(\psi(s_t; \theta_{\psi})), \quad \forall s \in \mathcal{S},$$

where $V_{\pi}(\psi(s_t; \theta_{\psi})) = \mathbb{E}_{\pi}\left[\sum_{t=0}^{\infty} \gamma^t r^{(e)}(x_t, a_t) \mid x_0 = x\right].$

**PPO**  PPO is a policy gradient method that improves stability by constraining policy updates. It maximizes the clipped surrogate objective:

$$L_{\pi}(\theta_{\pi}) = \mathbb{E}_t\left[\min(r_t(\theta_{\pi})A_t, \text{clip}(r_t(\theta_{\pi}), 1 - \epsilon, 1 + \epsilon)A_t)\right],$$

where $r_t(\theta_{\pi}) = \frac{\pi_{\theta}(\psi(s_t; \theta_{\psi}))}{\pi_{\theta_{\text{old}}}(a_t \mid \psi(s_t; \theta_{\psi}))}$ is the probability ratio and $A_t$ is the advantage function estimated via Generalized Advantage Estimation (GAE). The value function $V_{\phi}(\psi(s_t; \theta_{\psi}))$ is updated using:

$$L_V(\phi) = \mathbb{E}_t\left[(V_{\phi}(\psi(s_t; \theta_{\psi})) - V_t^{\text{target}})^2\right].$$

The final objective combines policy loss, value loss, and entropy regularization:

$$L(\theta_{\pi}, \phi_{\pi}) = L(\theta_{\pi}) - c_1 L_V(\phi) + c_2 H(\pi_{\theta}),$$

where $H(\pi_{\theta})$ encourages exploration.

**SAC**  SAC is an off-policy actor-critic method that optimizes not only for high expected returns but also for high-entropy policies, encouraging broader exploration.

The maximum-entropy RL objective is:

$$\max_{\pi} \sum_{t=0}^{\infty} \mathbb{E}_{s_t, a_t \sim \rho_{\pi}}\left[r^{(e)}(s_t, a_t) + \alpha\, \mathcal{H}\big(\pi(\cdot \mid s_t)\big)\right],$$

where $\mathcal{H}$ denotes the entropy of the policy, and $\alpha$ is a temperature parameter that balances exploration vs. exploitation. In practice, $\alpha$ can be fixed or automatically tuned.

SAC maintains two Q-functions, $Q_{\theta_1}$ and $Q_{\theta_2}$, to reduce positive bias in the target updates. Each $Q_{\theta_i}$ is learned by minimizing the mean-squared Bellman error:

$$L_{Q_i}(\theta_i) = \mathbb{E}_{(s_t, a_t) \sim D}\left[\big(Q_{\theta_i}\big(\psi(s_t; \theta_{\psi}), a_t\big) - y_t\big)^2\right],$$

where $D$ is the replay buffer, and

$$y_t = r^{(e)}(s_t, a_t) + \gamma\, \mathbb{E}_{s_{t+1} \sim p}\left[\min_{i=1,2} Q_{\theta_i^{\text{targ}}}\big(\psi(s_{t+1}; \theta_{\psi}), a_{t+1}\big) - \alpha \log \pi_{\phi}\big(a_{t+1} \mid \psi(s_{t+1}; \theta_{\psi})\big)\right].$$

Here, $Q_{\theta_i^{\text{targ}}}$ are target networks (periodically updated copies of $Q_{\theta_i}$), and $\pi_{\phi}$ is the current policy.

The policy $\pi_{\phi}$ is updated by minimizing the Kullback–Leibler divergence between $\pi_{\phi}$ and an exponential of the Q-function. In practice, the policy loss is often written as:

$$L_{\pi}(\phi) = \mathbb{E}_{s_t \sim D}\left[\mathbb{E}_{a_t \sim \pi_{\phi}(\cdot \mid \psi(s_t; \theta_{\psi}))}\left[\alpha \log \pi_{\phi}\big(a_t \mid \psi(s_t; \theta_{\psi})\big) - Q_{\theta_i}\big(\psi(s_t; \theta_{\psi}), a_t\big)\right]\right].$$

Because there are two Q-functions, the minimum of $Q_{\theta_1}$ and $Q_{\theta_2}$ is typically used in practice to reduce overestimation bias.

To automatically adjust the trade-off between exploration and exploitation, SAC can learn the temperature $\alpha$. The corresponding loss is:

$$L(\alpha) = \mathbb{E}_{s_t \sim D,\, a_t \sim \pi_\phi}\Big[-\alpha \, \log \pi_\phi\big(a_t \mid \psi(s_t; \theta_\psi)\big) - \alpha \, \bar{H}\Big],$$

where $\bar{H}$ is a target entropy. Decreasing $\alpha$ reduces the entropy bonus and focuses more on reward maximization, while increasing $\alpha$ encourages broader exploration.

SAC alternates between:

1. Minimizing the Q-function losses $L_{Q_1}(\theta_1)$ and $L_{Q_2}(\theta_2)$.

2. Minimizing the policy loss $L_\pi(\phi)$.

3. (Optionally) Adjusting $\alpha$ by minimizing $L(\alpha)$.

Thus, the SAC objective can be summarized as:

$$L_{\text{SAC}} = \sum_{i=1}^{2} L_{Q_i}(\theta_i) \;+\; L_\pi(\phi) \;+\; \begin{cases} L(\alpha) & (\text{if } \alpha \text{ is learned}), \\ 0 & (\text{otherwise}). \end{cases}$$

This encourages policies that achieve high returns while maintaining sufficient exploration through entropy maximization, all in the learned latent space $\psi(\cdot; \theta_\psi)$.

## F    Comparative Table Methodologies

| Feature | EMERALD | CaDM | PEARL |
|---|---|---|---|
| Context Conditioning | Implicit | Explicit | Explicit |
| Latent Encoding | State-space | Context | Context |
| Encoder model needed | No | Yes | Yes |
| Policy Input | Task-aware abstract state space $\mathcal{X}$ | $\mathcal{S}$ and $\mathcal{Z}$ | $\mathcal{S}$ and $\mathcal{Z}$ |
| Encoder model needed | No | Yes | Yes |
| Model Based or Model Free | Hybrid | Hybrid | Model-free |

Table 6: Feature Comparison Between Different Methods. $\mathcal{X}$ denotes a compressed representation of $\mathcal{S}$.

# G  Abstract Representation Model Architecture

We implement an abstract representation model designed for meta-reinforcement learning across a distribution of tasks $\mathcal{T}_i \sim p(\mathcal{T})$. The model maps observations $s_t \in \mathcal{S}$ and actions $a_t \in \mathcal{A}$ to latent abstract representations $x_t \in \mathcal{X}$, and infers task embeddings $z_i \in \mathcal{Z}$. Each component of the model is parametrized by submodules of $\theta$.

**State Encoder** $\psi_{\theta_\psi} : \mathcal{S} \times \mathcal{Z} \to \mathcal{X}$  The encoder $\psi$ maps observations $s_t$ and task embeddings $z_i$ to abstract states $x_t$. It includes skip connections to promote identity preservation and stabilize representation learning.

**Transition Function** $\tau_{\theta_\tau} : \mathcal{X} \times \mathcal{A} \to \mathcal{X}$  The transition model predicts the next abstract state $x_{t+1}$ given the current abstract state $x_t$ and action $a_t$. It is implemented as a deep MLP and is context-agnostic, relying on the encoded state $x_t$ to contain relevant task information.

**Reward Predictor** $\rho_{\theta_\rho} : \mathcal{X} \times \mathcal{A} \to \mathbb{R}$  The reward model estimates the immediate reward $r_t$ based on the abstract state and action.

**Task Encoder** $\phi_{\theta_\phi} : h_{i,:t} \to \mathcal{Z}$  An LSTM-based encoder $\phi$ that produces a task embedding $z_i \in \mathcal{Z}$ from the interaction history $h_{i,:t} = (s_0, a_0, r_1, s_1, a_1, r_2, \ldots, s_t)$. This embedding captures task-specific dynamics for each $\mathcal{T}_i$.

**Terminal Classifier** $\beta_\theta : \mathcal{X} \times \mathcal{A} \to \{0, 1\}$  A binary classifier that predicts episode termination given the abstract state and action. Note used in current setup, but available in the implementation.

**Discount Decoder**  An auxiliary module that estimates the discount factor $\gamma_t \in [0, 1]$ from $(x_t, a_t)$. This can be used during value prediction in planning.

**Inference Procedure**  At each time step $t$, the model operates as follows:

1. Compute the task embedding: $z_i = \phi(h_{i,:t})$

2. Encode current and next states: $x_t = \psi(s_t, z_i)$, $x_{t+1} = \psi(s_{t+1}, z_i)$

3. Predict the next abstract state: $\hat{x}_{t+1} = \tau(x_t, a_t)$

4. Predict the reward: $\hat{r}_t = \rho(x_t, a_t)$

5. Predict terminal status: $\hat{d}_t = \beta(x_t, a_t)$

The model is trained end-to-end using supervised objectives on transition, reward, and termination prediction. Training minimizes the expected loss across the task distribution:

$$\mathbb{E}_{\mathcal{T}_i \sim p(\mathcal{T})} \left[ \mathcal{L}_i(\theta) \right]$$

# H    Additional Experiments

**Structured Representation Learning for Structurally Similar MDPs**

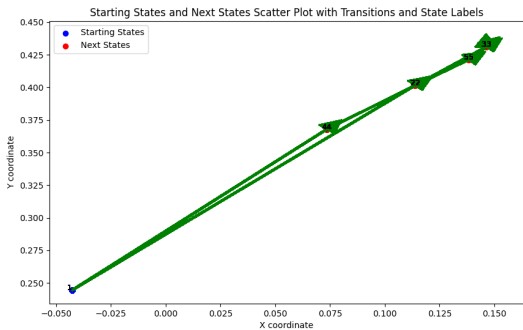

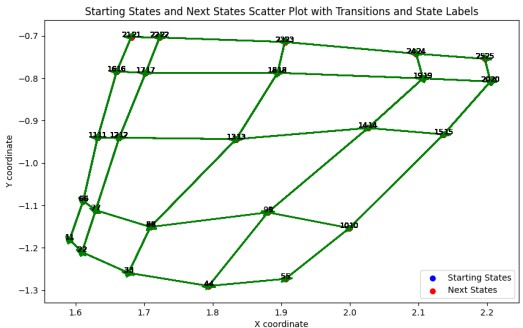

Figure 7: Reconstruction of an "arrow" from two MDPs.

Figure 8: Reconstruction of a maze-like structure from three different structural MDPs

To evaluate EMERALD's ability to produce meaningful and interpretable latent structures—a core property of abstract representation models—we adapt the visual experimental setups of Van Driessel & François-Lavet (2021) and François-Lavet et al. (2019) to a setting with multiple underlying MDPs (and thus multiple environments). Our objective is to assess how well EMERALD can learn a shared abstract representation across distinct environments.

We design two simple, interpretable MDP structures. In the first setting, we sample 300 transitions from two deterministic MDPs:

$$\mathcal{T}_1 : S_1 \to S_2 \to S_3, \quad \mathcal{T}_2 : S_1 \to S_4 \to S_2 \to S_5 \to S_3.$$

The transitions are embedded into a 2D latent space. As shown in Figure 7, EMERALD's ARM effectively captures the shared structure between the two environments in this space.

In the second setting, we consider three distinct 5×5 maze-like environments. For each maze, we collect 10,000 transitions under a random policy, generating random walks. The maze layouts are generated randomly, with a fixed start state at $(0, 0)$ and goal state at $(4, 4)$. A visual overview of the layouts is provided in Figure 9. In the resulting 2D latent representation (Figure 8), EMERALD successfully captures shared structural elements across the three mazes: similar states in different environments are closely aligned, illustrating the utility of the learned abstract representation.

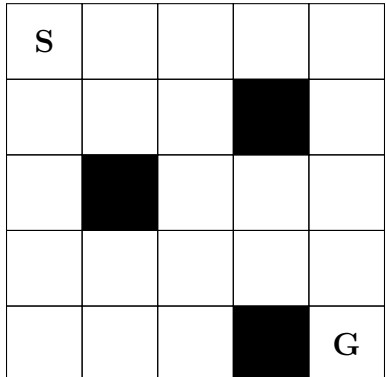
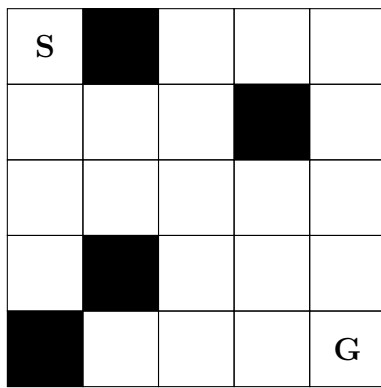
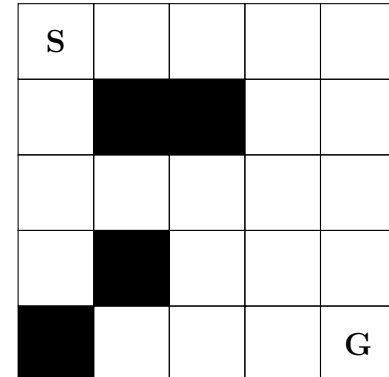

Figure 9: Three Maze Environments

**Effect of Input Policy Type on Performance**

For fairness, we pretrain the model with a random policy. However, to illustrate the effect when the offline data is of better quality (i.e. follows a non-random, but rather expert policy), we conducted a small ablation study with the PPO version of our model. The results are shown in Figure 7.

|  | Δ Improvement HalfCheetah (Vol.) | Δ Improvement HalfCheetah (Dir.) |
|---|---|---|
| EMERALD PPO | **214.6** ± 129.1 | **431.4** ± 161.9 |

Table 7: Performance improvement (Δ Mean Return ± Std.) of EMERALD PPO relative to baselines on HalfCheetah Volume and Direction tasks.

**Additional Meta-Word Suite Experiments**

Table 8: PEARL (SAC) baseline on Meta-World ML1 tasks.

| Method | door-open | basketball | window-open | pick-place | button-topdown |
|---|---|---|---|---|---|
| PEARL (SAC) | $0.0 \pm 0.0$ | $0.0 \pm 0.0$ | $36 \pm 1.1$ | $0.0 \pm 0.0$ | $43 \pm 2.5$ |

Table 9: PEARL (SAC) baseline on Meta-World ML10.

| Method | Average (Train) | Average (Test) |
|---|---|---|
| PEARL (SAC) | $23.1 \pm 1.5$ | $13.4 \pm 1.5$ |

# I Abstract Model Hyperparameters

Table 10: Training Model Parameters

| Parameter | HalfCheetah (Vol) | HalfCheetah (Dir) | Pendulum (Length) | CartPole (Length) |
|---|---|---|---|---|
| Weight Decay | 0.00001 | 0.00001 | 0.00001 | 0.00001 |
| Hidden Dim | 128 | 128 | 32 | 32 |
| Latent Dim | 10 | 10 | 2 | 8 |
| Model Learning Rate | 0.0001 | 0.0001 | 0.0001 | 0.0001 |
| Hidden Dim $\rho$ | 128 | 128 | 8 | 64 |
| Output Dim $\rho$ | 1 | 1 | 1 | 1 |
| Context Dim | 4 | 4 | 12 | 4 |
| Environment ID | HalfCheetah | HalfCheetah | Pendulum | CartPole |
| # Envs | 5 | 2 | 11 | 5 |

Table 11: Training Agent Parameters

| Parameter | HalfCheetah (Vol) | HalfCheetah (Dir) | Pendulum (Length) | CartPole (Length) |
|---|---|---|---|---|
| Policy Network LR | 0.0001 | 0.0001 | 0.0001 | 0.0003 |
| Policy Hidden Dim | 64 | 64 | 64 | 64 |
| Gamma | 0.95 | 0.95 | 0.99 | 0.99 |
| Reward Weight | 1 | 1 | 1 | 1 |
| $\tau$ Loss Weight | 1 | 1 | 1 | 1 |
| Regularization Weight | 0.01 | 0.01 | 0.01 | 0.01 |
| Train Epochs | 300 | 300 | 3000 | 50 |
| Max Iterations | 1000 | 1000 | 1000 | 200 |
| History Length | 100 | 100 | 100 | 100 |
| Max Episode Steps | 1000 | 1000 | 200 | 200 |
| Policy Iterations Learn | 5,000,000 | 5,000,000 | 2,000,00 | 2,000,00 |
| Samples per Task (ARM Training) | 100,000 | 100,000 | $\sim$4,500 | 10,000 |
| Batch Size | 256 | 256 | 256 | 256 |

## J  Training Details for the Different Environments

**General Details**  We follow the experimental setup of Lee et al. (2020), reproducing their results with the same environment modifications using the code base available at `https://github.com/younggyoseo/CaDM`. For each environment, we train on $N$ training configurations and evaluate on several out-of-distribution (OOD) test environments. Results are averaged over 8 runs with different random seeds. The setup is illustrated in Figure 10. While the conceptual design is the same, we modified the implementation to be compatible with OpenAI Gymnasium v5, which has built-in support for MuJoCo-based environments.

We implemented the EMERALD abstract representation model in PyTorch (Python 3.9). RL agents were implemented using Stable Baselines3 (SB3) (Raffin et al., 2021) and CleanRL (Huang et al., 2022). Experiments were conducted on an Apple M4 processor with 10 CPU cores and 10 GPU cores. All results are reported as the mean over 4 runs with different random seeds.

# K   Implementational Considerations

**OOD Comparative Experiments**   We sample offline data using a random policy to ensure that our model is not positively biased by policy optimization. This setup arguably places our model at a disadvantage compared to methods that benefit from task- or goal-directed data collection. We sample at most 1M training data points, divided by the number of maximum steps per episode and the number of available training environments. Full training details are provided in Table I. For baselines, we re-ran the publicly available code from `https://github.com/younggyoseo/CaDM` without any modifications.

**ID Comparative Experiments**   We compare in-distribution performance against existing methods. For VariBAD and PEARL, we report values as recorded in the original papers (Zintgraf et al., 2021).

**Task-Diversity Experiment**   For each environment, we train models (ARMs) under two configurations, averaging results over five runs, while keeping the total number of environment interactions fixed: 200K for CartPole and 3M for HalfCheetah. In Configuration 1, each model is trained on a single task using the full sample budget. In Configuration 2, the same total number of samples is distributed across five tasks—i.e., 200K / 5 for CartPole and 3M / 5 for HalfCheetah—reducing per-task data while increasing task diversity.

**Ablation Study**   We evaluate the model under four different configurations:

1. 100K samples with a context encoder on CartPole (Volume).

2. 10K samples without a context encoder on CartPole.

3. 100K samples with a context encoder on HalfCheetah (Volume).

4. 10K samples without a context encoder on CartPole.

We trained each configuration for up to 100 epochs to analyze the effect of model training on downstream performance. After pretraining, we evaluate by fine-tuning the policy on a single target environment: CartPole with pole length 1, or HalfCheetah (Volume) with scaling set to 1. We fine-tune for 10K steps in CartPole and 100K steps in HalfCheetah. We deliberately keep fine-tuning short to avoid masking the effects of model pretraining—since longer training might allow some seeds to recover good performance regardless of initialization, thus obscuring the contribution of the model.

**Latent Trajectory Analysis**   For each environment, we extract the first 50 latent transitions from both training and test episodes. In CartPole and HalfCheetah (Volume), this yields 7 distinct clusters (5 from training environments and 2 from test environments), while in Pendulum (Length), we obtain 13 clusters (11 training, 2 test). By design, the initial training transitions are collected from largely untrained policies, resulting in greater variability within the corresponding latent clusters. In contrast, test clusters reflect more stable behavior, as they are derived from trained policies. To visualize the latent space structure, we apply principal component analysis (PCA) to reduce the dimensionality and plot the trajectories in 3D.

This form of latent space analysis has been used in prior work to examine representation quality and clustering behavior in RL and meta-RL models (e.g., Rakelly et al., 2019; Zintgraf et al., 2021).

## L Environment-Variations

We follow the general structure of Lee et al. (2020).

**CartPole**   The CartPole task involves balancing a pole on a moving cart by applying discrete forces to the cart.

- **Observation:** $(x_t, \dot{x}_t, \theta_t, \dot{\theta}_t)$, where $x$ is the cart's position and $\theta$ is the angle of the pole from the vertical.

- **Action:** $\{0, 1\}$, where 0 pushes the cart left and 1 pushes it right.

- **Reward:**
$$r_t = \mathbb{1}_{\{|x_{t+1}|<2.4 \wedge |\theta_{t+1}| < \frac{14\pi}{360}\}}$$
A reward of 1 is given if the cart remains within $\pm 2.4$ and the pole within $\pm 14°$; otherwise 0.

- **Modifications:** Push force $f$ and pole length $l$ are modifiable.

**Pendulum**   The Pendulum task aims to swing a rod upright using continuous torque.

- **Observation:** $(\cos\theta, \sin\theta, \dot{\theta})$, where $\theta \in [-\pi, \pi]$, with $\theta = 0$ being upright.

- **Action:** $a \in [-2.0, 2.0]$, representing continuous torque.

- **Reward:**
$$r_t = -\left(\theta_t^2 + 0.1\dot{\theta}_t^2 + 0.001 a_t^2\right)$$
Penalizes deviation from the upright, angular velocity, and torque magnitude.

- **Modifications:** Pendulum mass $m$ and length $l$ can be changed.

**HalfCheetah (Volume)**   The HalfCheetah is a planar robot designed to learn fast and energy-efficient running.

- **Observation:** A 20-dimensional vector including joint positions/velocities, root joint state (excluding x), and torso center-of-mass velocity.

- **Action:** $a \in [-1.0, 1.0]^6$, torques applied at six joints.

- **Reward:**
$$r_t = \dot{x}_{\text{torso},t} - 0.05\|a_t\|^2$$
Rewards forward velocity and penalizes control effort.

- **Modifications:** Scale the mass of every rigid link by a fixed factor $m$ and the damping of every joint by a fixed factor $d$.

**HalfCheetah (Direction)**   HalfCheetah is a planar robot with 9 links and 6 actuators, designed to move along the x-axis. The agent is trained to move either forward or backward depending on the task direction.

- **Observation:** A 17-dimensional vector containing positional and velocity information of the joints and torso.

- **Action:** $a \in [-1, 1]^6$, representing the torques applied at each joint.

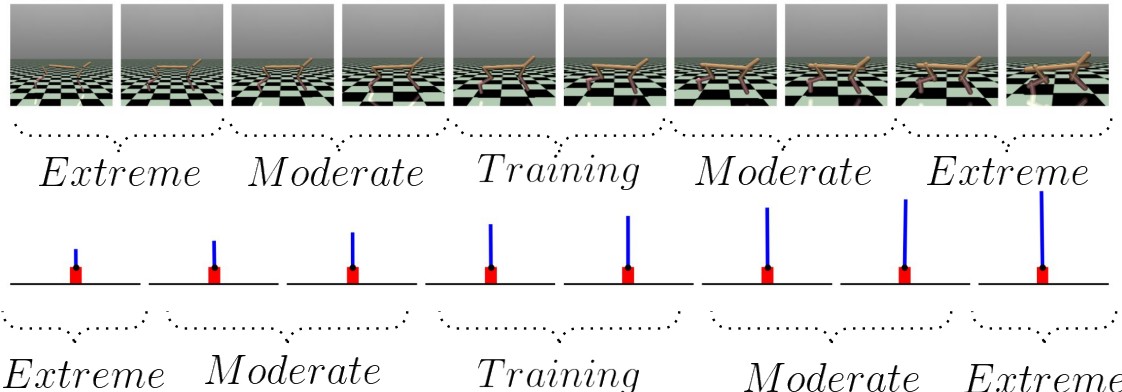

Figure 10: Illustration of the Experimental Setup. We train on $N$ intermediate training environments. At test time, we test the model's performance in a test regime that is moderately different and one that differs extremely from the training environment.

- **Reward:**

$$r_t = d \cdot \dot{x}_t - 0.05\|a_t\|^2$$

    where $d \in \{-1, 1\}$ specifies the target direction (backward or forward). The reward encourages movement in the desired direction and penalizes large control inputs.

- **Modifications:**

    1. Scale the mass of each link by a factor $m$
    2. Scale joint damping by a factor $d$

Table 12: Environment Modifications used for Comparative our Experiments (following Lee et al. (2020)).

| Environment | Training | Test (Moderate) | Test (Extreme) | Episode Length |
|---|---|---|---|---|
| CartPole | $f \in \{5.0, 6.0, 7.0, 8.0, 9.0, 10.0$ $11.0, 12.0, 13.0, 14.0, 15.0\}$ $l \in \{0.40, 0.45, 0.50, 0.55, 0.60\}$ | $f \in \{3.0, 3.5, 16.5, 17.0\}$ $l \in \{0.25, 0.30, 0.70, 0.75\}$ | $f \in \{2.0, 2.5, 17.5, 18.0\}$ $l \in \{0.15, 0.20, 0.80, 0.85\}$ | 200 |
| Pendulum | $m \in \{0.75, 0.80, 0.85, 0.90, 0.95,$ $1.00, 1.05, 1.10, 1.15, 1.20, 1.25\}$ $l \in \{0.75, 0.80, 0.85, 0.90, 0.95,$ $1.0, 1.05, 1.10, 1.15, 1.20, 1.25\}$ | $m \in \{0.50, 0.70, 1.30, 1.50\}$ $l \in \{0.50, 0.70, 1.30, 1.50\}$ | $m \in \{0.20, 0.40, 1.60, 1.80\}$ $l \in \{0.20, 0.40, 1.60, 1.80\}$ | 200 |
| HalfCheetah (Volume) | $m \in \{0.75, 0.85, 1.00, 1.15, 1.25\}$ $d \in \{0.75, 0.85, 1.00, 1.15, 1.25\}$ | $m \in \{0.40, 0.50, 1.50, 1.60\}$ $d \in \{0.40, 0.50, 1.50, 1.60\}$ | $m \in \{0.20, 0.30, 1.70, 1.80\}$ $d \in \{0.20, 0.30, 1.70, 1.80\}$ | 1000 |
| HalfCheetah (Direction) | $d \in \{-1.0, 1.0\}$ | $d \in \{-1.0, 1.0\}$ | $d \in \{-1.0, 1.0\}$ | 1000 |

## M    Budget Allocation

| Environment | Total Number of Transitions | ARM Training | Policy Training |
|---|---|---|---|
| HalfCheetah | 5M | 500K | 4.5M |
| Pendulum | 500K | ∼50K | ∼450K |
| Cartpole | 500K | 50K | 450K |

Table 13: Sample breakdown across training phases for each environment.

# N    Additional Considerations regarding Baselines

**Hyperparameter Tuning**: We aimed to reproduce the baselines as faithfully as possible by relying directly on the authors' publicly available implementations. For each method, we used the original source code for the networks and policies whenever feasible. To ensure fairness in comparison, we applied the same hyperparameter tuning protocol to both our method and the baselines. Specifically, for every method we selected the best-performing discount factor $\lambda \in 0.9, 0.95, 0.99$ and learning rate $\in 10^{-3}, 10^{-4}, 10^{-5}$.

**Observed Discrepancies**: We also verified the extent to which our experimental results deviate from existing literature. For Setting 1, our findings largely align with prior work (most notably Lee et al. (2020)). The same holds for the HalfCheetah (Direction) experiment in Table 2, although we were only able to validate the VariBAD (PPO) results using Zintgraf et al. (2021) and could not find any reported performance for VariBAD (SAC) in this setting. In Settings 2 and 3, the PPO-based results are generally consistent with the values reported in the original papers (e.g., Beck et al. (2023); Ni et al. (2021)). A notable exception is the VI-HN (PPO) result from Beck et al. (2023), which in our experiments performs substantially better than RNN-HN. Finally, most hypernetwork-based SAC approaches perform significantly worse than their PPO counterparts in the MetaWorld settings. We could not find any prior literature demonstrating strong SAC performance in these settings, which may indicate that others have attempted SAC without success.

**Additional Considerations**: We further highlight the following implementational challenges: (1) It was not always possible to run the source code *as is*; in several cases we had to modify or debug the implementations to make them functional. This was particularly true for RMF-RL Ni et al. (2021) and RNN-HN Beck et al. (2023). (2) Another difficulty and potential source of variation stems from the need to adapt the baselines to our Gymnasium environment version (1.1.1). Most original implementations rely on earlier versions of Gym and/or MuJoCo. Nevertheless, our VI-HN (PPO) and RNN-HN (PPO) results closely match those reported in Beck et al. (2023) (see Section 8). (3) Finally, we maintain a strict distinction between PPO-based and SAC-based methods. However, many official repositories provide a complete implementation for only one of these families (usually only the on-policy PPO). For example, the original VariBAD codebase (`https://github.com/lmzintgraf/varibad`) includes PPO and A2C implementations but no SAC version. For this reason, we implemented most of the SAC variants ourselves. While we managed to do so within our available time and resources, we emphasize that integrating SAC into PPO-oriented codebases could likely be improved, though doing so would require methodological design changes beyond the scope of this work. Still, we did test whether SAC-based approaches such as VI-HN (SAC) worked in single-setting problems and found that we obtained reasonable results there, indicating that there are almost certainly no large bugs in the code and that breakdowns only occur when the method needs to generalize to the meta-RL setting.

## O   Averaged Performance for SAC and PPO

The table below show the avrage performance over *both* PPO and SAC.

| | CartPole (Lengths) | | Pendulum (Lengths) | | HalfCheetah (Volume) | |
|---|---|---|---|---|---|---|
| | Moderate | Extreme | Moderate | Extreme | Moderate | Extreme |
| VariBAD (Avg.) | 196.9 | 190.8 | -659.9 | -918.5 | 2359.8 | 1521.6 |
| RMF-RL (Avg.) | 198.35 | 194.55 | -637.45 | -978.9 | 3361.3 | 1589.6 |
| VI-HN (Avg.) | **198.45** | 195.35 | -539.25 | -814.95 | 3123.75 | 1956.1 |
| RNN-HN (Avg.) | 197.5 | 194.2 | -591.65 | -744.55 | 2410.3 | 1761.75 |
| EMERALD (Avg.) | 196.85 | **197.75** | **-255.35** | **-531.55** | **4021.8** | **2336.6** |

Table 14: Averaged performance between PPO and SAC versions (Mean Return).

# P  Learning Curves

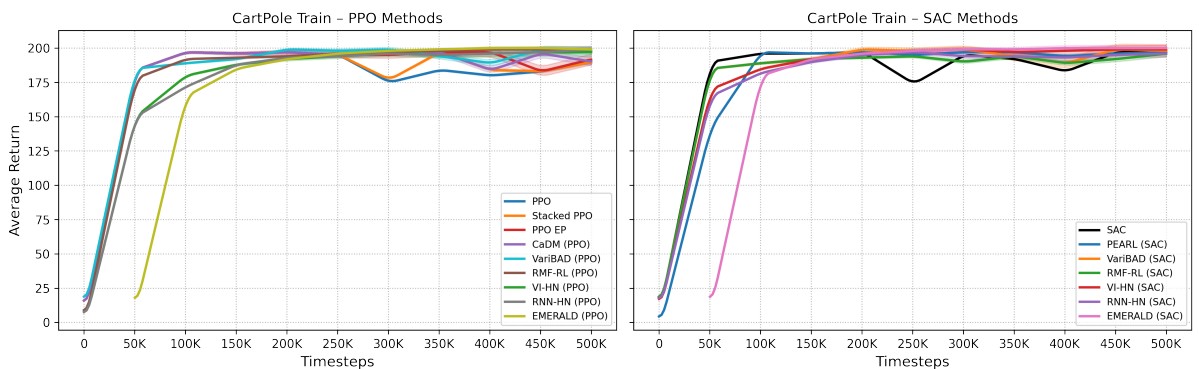

Figure 11: CartPole: Training (rolling averages over 8 independent runs)

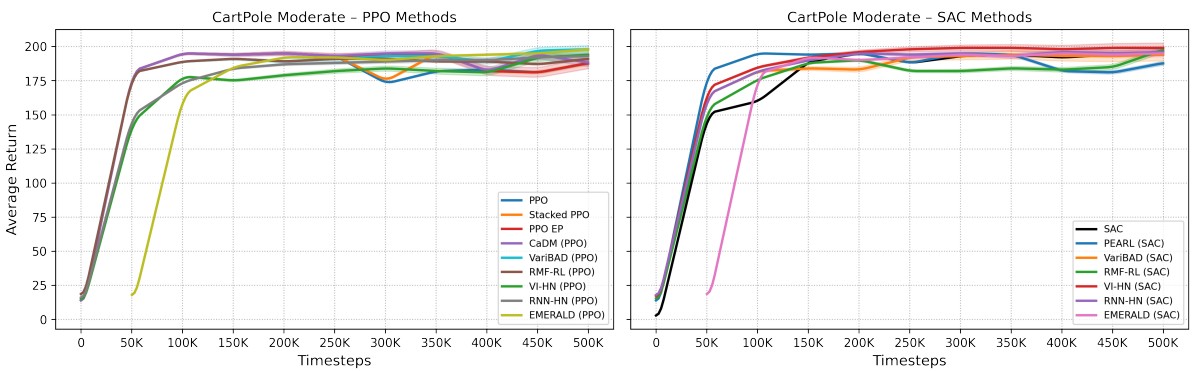

Figure 12: CartPole: Moderate (rolling averages over 8 independent runs)

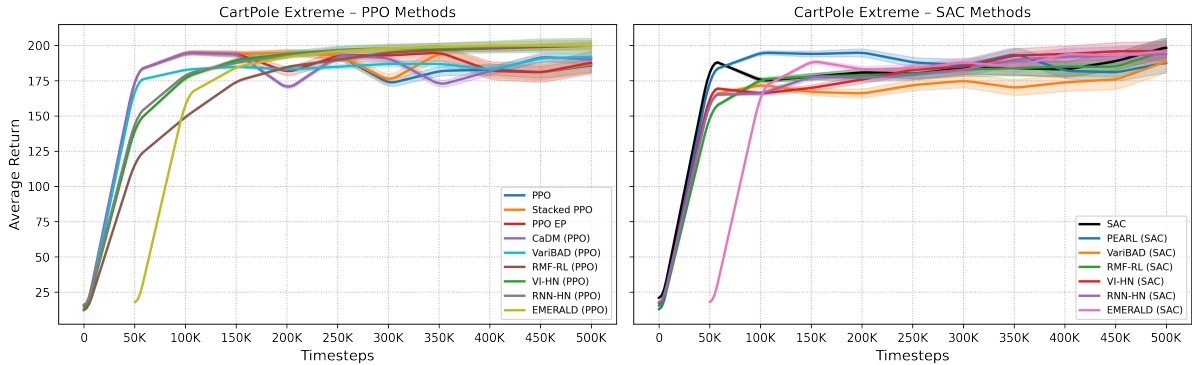

Figure 13: CartPole: Extreme (rolling averages over 8 independent runs)

Performance on Pendulum Train

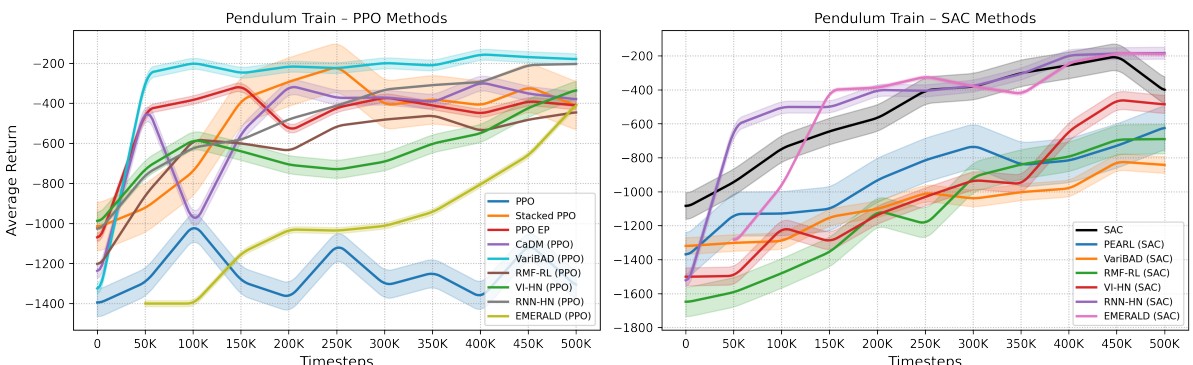

Figure 14: Pendulum: Training (rolling averages over 8 independent runs)

Performance on Pendulum Moderate

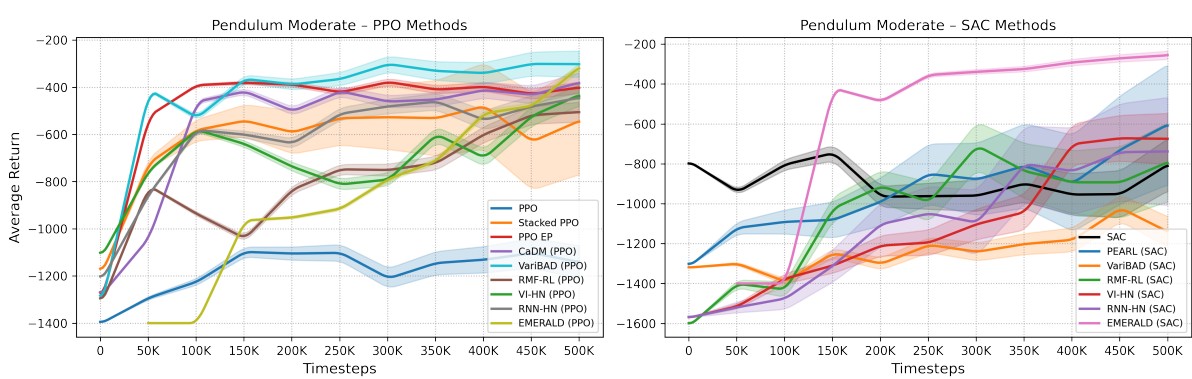

Figure 15: Pendulum: Moderate (rolling averages over 8 independent runs)

Performance on Pendulum Extreme

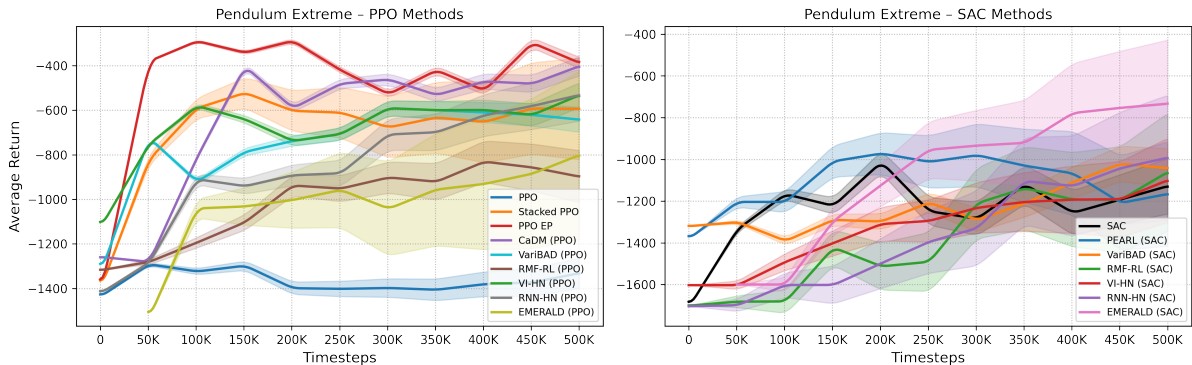

Figure 16: Pendulum: Extreme (rolling averages over 8 independent runs)

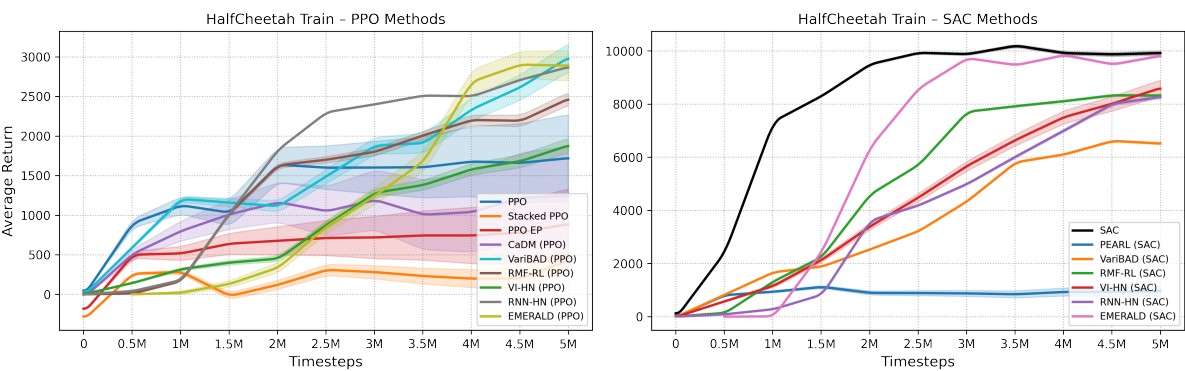

Figure 17: HalfCheetah (Volume): Training (rolling averages over 8 independent runs)

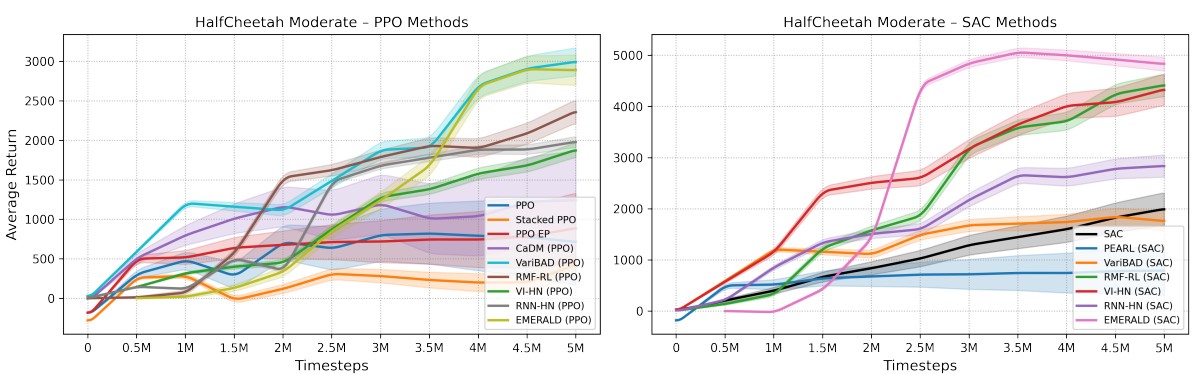

Figure 18: HalfCheetah (Volume): Moderate (rolling averages over 8 independent runs)

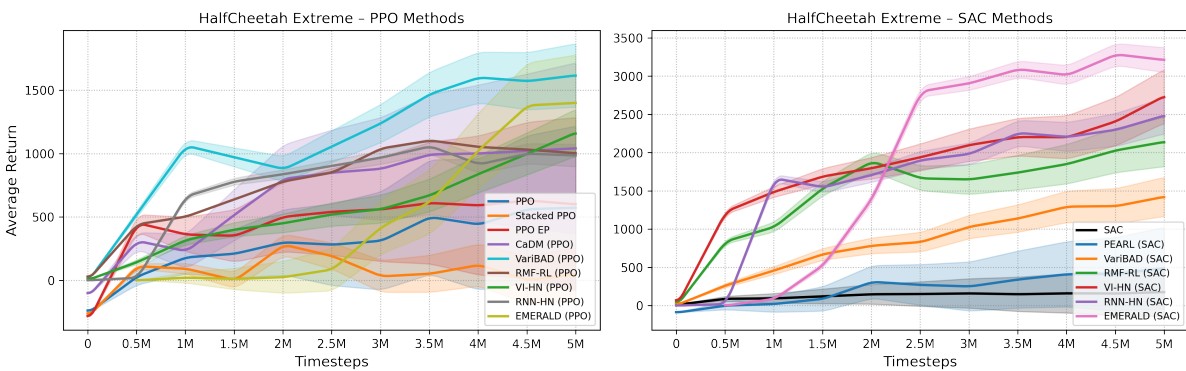

Figure 19: HalfCheetah (Volume): Extreme (rolling averages over 8 independent runs)

# Q    Latent Space Visualizations

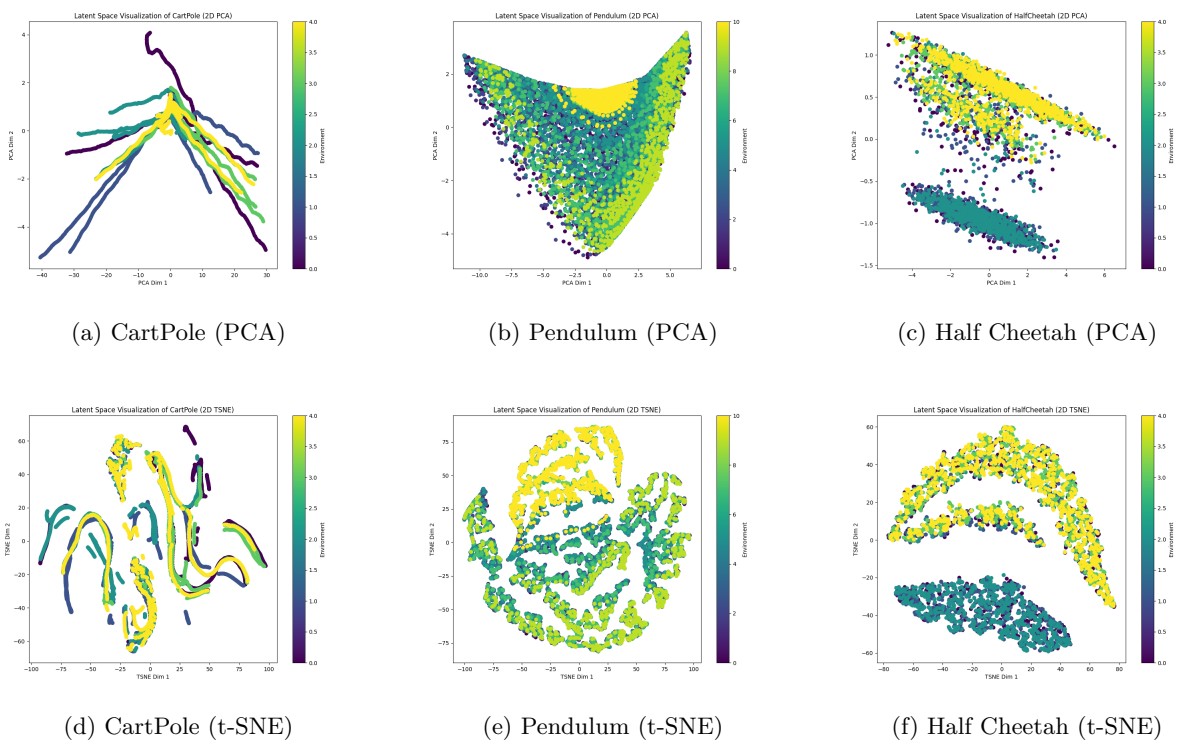

(a) CartPole (PCA)     (b) Pendulum (PCA)     (c) Half Cheetah (PCA)

(d) CartPole (t-SNE)    (e) Pendulum (t-SNE)    (f) Half Cheetah (t-SNE)

Figure 20: Latent space visualizations across environments using PCA (top row) and t-SNE (bottom row).

