# OpenReview forum: "Learning Task-Aware Abstract Representations for Meta-Reinforcement Learning"
_TMLR — Accepted by TMLR_

### Review · Reviewer_99cF · 2025-09-23

**Summary Of Contributions:**

The paper proposes a new framework for meta-reinforcement learning (meta-RL), which consists in learning a task-aware abstract state representation that encodes information both about the state and about the task at hand. This is achieved by conditioning the abstract state representation not only on the current state but also on the MDP trajectory, since the latter conveys critical information about the task. In order to ensure that the discovered abstract representation is meaningful, the authors introduce several loss terms that capture various aspects of the MDP, such as the transition and reward functions, while ensuring that the learned representations are non-trivial (i.e., avoiding representation collapse). The framework is generic enough to be used with any off-the-shelf RL algorithm (such as PPO or SAC). The authors conduct an experimental study with standard meta-RL benchmarks to demonstrate the performance of their framework.

**Audience:**

Yes

**Audience Explanation:**

Meta-RL is a significant research area with several real-world applications. Furthermore, the paper explains how it is related to the existing RL literature and how it tries to advance it. In that sense, I feel the work is definitely relevant to a subset of the TMLR audience.

**Broader Impact Concerns:**

I have no broader impact concerns.

**Claims And Evidence:**

No

**Claims Explanation:**

The paper proposes an interesting framework, but I have several concerns regarding a number of points:
- Is the framework totally model-free? The authors implicitly learn the transition and reward functions, as part of training the abstract state representation model. In Algorithm 2, the authors opt for policy learning, but given the framework also learns a state transition model as well as a reward model, the approach could in principle also be combined with model-based RL methods? I was unclear as to how the authors position their framework, but I feel it has strong elements of model-based approaches (especially as far as Algorithm 1 is concerned).
- Even though I understand where the authors where the authors are trying to get at with Proposition 1,  found the exposition quite confusing and possibly not so rigorous. First, the authors assume a task embedding $z$ of the form $z=\phi(h_{1:t})$ in the statement of Proposition 1. Under such an assumption, where the embedding depends on $h_{1:t}$, we expect that the mutual information between the input $h_{1:t}$ and the embedding $z$ will be strictly positive, unless we are in some weird pathological case. Actually, $h_{1:t}$ contains state $s_t$, and at the very least we expect $z$ to depend on $s_t$. So, I do not see how the mutual information term in Proposition 1 could be zero. Even if $z$ does not depend on the history leading up to $s_t$, it should at least depend on $s_t$ (which is contained in $h_{1:t}$). So, unless we are in some strange pathological scenario, the mutual information is expected to be positive. But even putting that aside, the proof is more of a sketch. A rigorous proof, for instance, would assume some embedding for case 1, and would show how modifying this embedding for the pair $(s_t,a_t)$ with different next state $Ss_{t+1}$ would result in a strictly lower loss. And I'm not even sure a separate proposition is needed. Perhaps the authors could just provide some intuitive arguments as to why conditioning on the task history is meaningful, without introducing extra propositions.
- I also found the proof in Lemma 2 confusing. In part (2) of Proof 3, the authors claim that the encoder cannot differentiate between tasks, leading to a collapsed latent representation. But then they go on to claim that the mapping $s_t\to x_t$ is one-to-many (not many-to-one). So, I think they need the opposite claim, i.e., that the mapping may actually map a given state $s_t$ to many different $x_t$ (the opposite of collapse. Again, I feel the proofs are not sufficiently rigorous and mostly resemble proof sketches.
- In Proposition 2, could the authors detail their derivation? $\psi(s_t,z)$ essentially corresponds to one random variable (that depends on two other random variables). But the term $\psi(s_t,z)$ itself is not two random variables. Could the authors then explain how they applied the general entropy inequality?
- Regarding the experimental evaluation, I'm not sure I was able to fully comprehend the results. First, it seems that there is so much variability depending on whether SAC or PPO is employed. I understand that some difference is to be expected, but it was very extreme. This was true in both the vanilla variants as well as the EMERALD variants. I'm not sure if the authors had any idea on that, but it would be nice to shed light on this. The reason why I believe this to be important is that what the authors ideally need to show is that the superior results are due to their framework, not due to the employed RL algorithm. To be more specific, if we look for example at HalfCheetah in Table 1, VariBAD in fact performs almost identically to EMERALD PPO in the moderate setting, and in fact outperforms EMERALD PPO in the extreme setting. It is of course true that EMERALD SAC beats VariBAD by a large margin in both settings. But, given the very large variability, it is not at the end of the day clear what causes the performance lift: is it the new framework or the RL algorithm? Notice that similar observations are true for CartPole and Pendulum.
- To make things worse, in Table 2 Emerald PPO performs similar to VariBAD, but this time EMERALD SAC exhibits very bad performance. What causes such a huge degradation from Table 1 to Table 2 for EMERALD SAC in the HalfCheetah environment? I understand one is out-of-distribution testing and the other is in-distribution setting, but the swing is too wild. I was unable to make sense of these results.
- I was unclear what Table 3 was trying to measure. Is the measured performance on only the tasks that the model was trained on? Focus for example on CartPole.  Config 1 has 1 training task, while Config 2 has 5. But is the mean return measured on 1 and 5 tasks, respectively, or on 5 tasks for both configurations? If it is the latter, then of course having more tasks should boost performance.
- I was unable to follow the arguments in Section 5.3 regarding Figure 5. The authors state that "In the CartPole setting with 10k transitions, returns plateau after roughly 10 epochs". Looking at Figure 5, how is that the case? Performance keeps increasing as we increase the number of epochs beyond 10 or even 60. I have no idea how Figure 5 shows a plateau. Similarly, the authors claim "Although the 100k-transition regime converges earlier, it does not exceed the final performance of the 10k model". But in HalfCheetah in Figure 5, task-aware performance in the 100k-transition regime is certainly higher than than in the 10k regime. It is as if the authors were referring to an entirely different figure.

**Requested Changes:**

I'd appreciate it if the authors could address all points raised above with proper clarifications. I would also have some additional questions:
- Have the authors tried more diverse tasks? It is not clear whether the proposed framework could handle for instance situations where the task distribution is much more diverse.
- How did the authors ensure that the performance comparison was fair? The authors mention in the Appendix that "For VariBAD and PEARL, we report values as recorded in the original papers". Was this the case for both OOD and ID settings? I totally understand the difficulty of implementing all methods and conducting all experiments, but then one must also ensure that the comparison is fair and meaningful.
- What was the purpose of the in-distribution experiments? In practice, aren't these algorithms expected to be used predominantly in out-of-distribution settings? Do we even need meta-RL for the in-distribution setting? One might claim that conventional RL can already deal with this case, assuming we have trained on all tasks?

---

> ### Author Response · Authors · 2025-10-19
> **Reviewer 99cF Comment from Authors**
>
> We first would like to thank the reviewer for the extensive feedback on our work. We are happy to hear they found the framework interesting.
>
> We will now continue with a clarification of the points of concern:
>
> > Is the framework totally model-free?
>
> - **Model-free vs. model-based:** We position EMERALD as a *model-free* method which effectively uses a pretrained encoder.
>
> > Even though I understand where the authors where the authors are trying to get at with Proposition 1, found the exposition quite confusing and possibly not so rigorous.
>
> - **Confusion regarding Prop. 1:** Thank you for this detailed observation. We refer to our global comment above.
>
> > I also found the proof in Lemma 2 confusing.
>
> **Confusion regarding Lemma 2:** Indeed, this is a good catch. We refer to the improved proposition and proof in our reply to Reviewer aQ3c.
>
> > In Proposition 2, could the authors detail their derivation?
>
> - **Derivation in Proof**: Please see the updated proof in our comment to Reviewer aQ3c, there you will find the complete (updated) derivation.
>
> > Regarding the experimental evaluation, I'm not sure I was able to fully comprehend the results.
>
> **General questions on experimental evaluations**: Our intention was to highlight that the ARM provides a consistent performance boost over each vanilla baseline. For example, PPO+EMERALD improves by $\sim$2100 points over Vanilla PPO in HalfCheetah (moderate), and SAC+EMERALD improves by $\sim$3000 points over Vanilla SAC. These consistent gains illustrate that the ARM substantially enhances base RL algorithms. To further illustrate competitiveness to other methods, we included some more experiments and baselines (see: Global Comment).
>
> > in Table 2 Emerald PPO performs similar to VariBAD, but this time EMERALD SAC exhibits very bad performance. What causes such a huge degradation from Table 1
>
> **Concerns regarding Table 2:** Table 2 does not simply present in-distribution results of the Table 1 setting; it corresponds to a different task. In Table 1, the HalfCheetah’s *dynamics* are modified by changing its volume, whereas in Table 2 only the reward function is modified based on context. We clarified this distinction in the revision, as the aim of Table 2 is to study performance under changing reward dynamics rather than environment dynamics.
>
> > I was unclear what Table 3 was trying to measure.
>
> **Clarification regarding Table 3**: Table 3 is intended to show that incorporating an ARM can boost performance. The mean is computed over 1 task in the first case and 5 tasks in the second, not 5 tasks in both configurations. We made this explicit in the revision.
>
> > I was unable to follow the arguments in Section 5.3 regarding Figure 5.
>
> **Arguments in Section 5.3:** You are correct that the current description is inaccurate (likely introduced by a version control error). Our intended point was that task-aware encodings consistently outperform task-agnostic ones across data regimes (CartPole and HalfCheetah), with especially pronounced differences in HalfCheetah (Volume).The aim is to show that removing task encodings degrades performance. We corrected the text and ensured Figure 5 reflects this properly in the revision. Our revision:
>
>  > We study how performance varies with (i) the number of training epochs, (ii) the amount of offline data, and (iii) the presence of task-aware encodings. Figure 5 reports the results (each marker is the mean of 10 seeds). In the CartPole setting with 10k transitions, we observe a slightly higher performance of the task-aware variant. Similarly, for CartPole with 100k transitions, the task-aware variant outperforms the task-agnostic one. In the HalfCheetah (Volume) setting, the same pattern emerges but is more pronounced: both the task-aware100k and 10k-transition regimes converge earlier. Moreover, the performance of the task-agnostic (10k) variant is effectively zero, which suggests that removing task encodings (i.e., using task-agnostic abstract states) sharply degrades performance, particularly on HalfCheetah, demonstrating the effectiveness of using a state-task encoder.
>
> # Requested Changes
>
> > Have the authors tried more diverse tasks?
>
> - **More diverse tasks**: The extreme settings already cover wide variations (e.g., Pendulum length 0.2–1.8), and additional Meta-World experiments further demonstrate the framework’s robustness to diverse tasks (see: additional experiments in Global Comment).
>
> > How did the authors ensure that the performance comparison was fair?
>
> - **Fairness of comparison**: All methods were re-implemented and run under identical conditions; for VariBAD, we report the original published value due to reproduction discrepancies. We clarified this point in the revision.
>
> - **Purpose of in-distribution tasks**: ID tasks demonstrate that meta-RL methods (e.g., VariBAD, PEARL, EMERALD) outperform standard RL baselines and that the ARM abstraction further enhances efficiency and performance.

---

### Review · Reviewer_aQ3c · 2025-10-03

**Summary Of Contributions:**

The paper proposes a task-inference method for meta-RL that makes use of trajectory reconstruction to learn the latent representation. In order to improve out-of-distribution generalization, the method proposes that 1) states and tasks be integrated in the model architecture before other inputs (e.g. actions) and that 2) the reconstruction loss predicts transitions over these "abstract" states (similar to a belief state in a POMDP). The paper adds an additional objective to prevent representation collapse and trains the encoder upfront on purely random data. The method is motivated by two theoretical claims and is evaluated on a MuJoCo task, Cartpole, and Pendulum. The key strengths of the paper are its fair comparisons (e.g. starting their method accounting for the data collection and using a random policy), its theoretical motivation, and the number of baselines. I have some concerns about the choice of baselines, interpretation and potential inaccuracy of the theoretical results, and reliance on a random policy, which are elaborated in the rest of the review.

**Audience:**

Yes

**Audience Explanation:**

Meta-RL is a popular field and these results are certainly relevant. The improvement on the HalfCheetah domain is substantial (although I do wonder how much of the improvement is coming from SAC vs the EMERALD method, and baselines could be stronger). Overall, I would certainly say that some individuals in TMLR would be interested.

**Claims And Evidence:**

Yes

**Claims Explanation:**

The claims are mostly accurate, with notable concerns listed here.
1. Proposition 1 seems to motivate the fact that sufficient statistics for the meta-RL problem must have a non-zero amount of information about history. It is already known that a history-dependent policy is necessary for such problems (Beck et al., 2025), and this observation does not seem to add any motivation for this method over the baselines. Is there another point to this proposition?
2. Proof 2 may have a mistake. I am not entirely certain, and would like clarification. A) In proof two, you cannot directly apply the subadditivity inequality H(X,Z) < H(X)+H(Z) because you are comparing H(\psi(s,z)) to H(\psi*(s)) + H(z), not H(s,z) to H(s) + H(z). It seems that the point of Lemma 1 and 2 is that \psi preserves entropy (if one-to-one, Lemma 2), whereas \psi*, as a stochastic one-to-many mapping, may increase entropy. But unless I am missing something, Lemma 1 only applies when the input variables are the same. In this case, aren’t the inputs to \psi and \psi* different? Is your point that H(\psi(s,z)) = H(s,z) (since ψ is one-to-one) and that H(\psi*(s)) can exceed H(s) because ψ* is stochastic? If so, then the argument that a one-to-many mapping has “higher entropy” is only valid for conditional entropy H[Y|X]: stochastic mappings increase the uncertainty of the output given the input. However, this does not imply that the marginal entropy H(Y) is larger; adding noise can increase or decrease H(Y) depending on how the noise interacts with the input distribution. Since your proof seems to compare marginals like H(\psi(s,z)) and H(\psi*(s)), I’m not sure the conditional-entropy argument applies directly. Could you clarify exactly where conditional vs. marginal entropy is being used, and how you are applying Lemma 1 and 2? B) Moreover, isn’t the more relevant comparison H(\psi(s,z)) to H(\psi*(s), z), since in the architecture you concatenate the task representation with z? Comparing to H(\psi*(s)) + H(z) only gives a loose upper bound, whereas the joint entropy H(\psi*(s), z) directly reflects the representation actually used. If this is the intended argument, could you clarify why you introduce the looser sum bound rather than comparing the joint entropies?
3. The baselines all seem to be other task-inference methods. It would really add a lot to compare to methods trained end-to-end on the policy objective, since those have recently been shown to achieve comparable or better performance -- Recurrent Model-Free RL Can Be a Strong Baseline for Many POMDPs (Ni et al., 2022) -- and some do so by encoding the state in a way dependent on the task -- Recurrent Hypernetworks are Surprisingly Strong in Meta-RL (Beck et al. 2023). Within the task-inference literature itself, it would be nice to compare to methods that use multi-task (pre-)training to learn only the relevant information from the task — e.g., Learning adaptive exploration strategies in dynamic environments through informed policy regularization (Kamienny et al., 2020) and Belief Agent (Humplik et al., 2019).
4. It is unclear which methods, other than Vanilla SAC, use SAC, and it is unclear what is meant by "Vanilla PPO" and "Vanilla SAC". PPO and SAC, to me, only specify the outer-loop. Do these methods use an RNN? In what ways are they similar to, and different from, EMERALD?

Less critical, but other issues that detract from a convincing paper:
5. Less critical, but a separate ablation for the architectural design (encoding state and task before the action) and the latent state reconstruction loss (as opposed to on the MDP state), would help to disentangle the contributions of each.
6. I am not convinced that using random policy data to train the task-inference module will generalize to other domains, and it also may account for the difference in results from VariBAD. For example, even in some toy T-Maze environments, a significant amount of exploration is needed before seeing a transition that would distinguish the tasks. More difficult baselines (e.g., Meta-World or manually designed) would help make this more convincing. At the same time, I wonder whether training on a random policy, on the domains where it is effective, improves transfer, and it would be nice to see a version of VariBAD that likewise trains on only offline data.

**Requested Changes:**

Critical for acceptance:
1. As mentioned above, proof 2 may have a mistake. If I am not incorrect about this mistake, then it would require correction. Note: I only checked proof 2, and not proof 1, in the appendix.

Borderline:
2. Could you clarify the motivation for proof 1 over the known fact that history is required to know the task in meta-RL?
3. Could you clarify Vanilla PPO and SAC?
4. Could you add end-to-end baselines (if this is not accounted for in Vanilla PPO and SAC)

Would strengthen the paper:
4. Task-inference baselines with multi-task pre-training.
5. Separate ablations for the architecture and objectives, as mentioned above.
6. More challenging environments.
7. VariBAD trained on random offline data.

---

> ### Author Response · Authors · 2025-10-19
> **Reviewer aQ3c Comment from Authors**
>
> We thank you for taking the time to carefully assess our work and are pleased to hear that the reviewer appreciates our theoretical motivation and comparisons to related methods!
>
> We now continue with addressing your main concerns:
>
> > Proposition 1[...] Is there another point to this proposition?
>
> - **Relevance of Prop. 1**: We included it as a formal, illustrative formalization (see also: comment regarding Proposition 1 above).
>
> > Proof 2 may have a mistake.
>
> - **Proof of Prop 2.**: Thank you for your insightful comments! Based on those, we propose a revision of the proposition and corresponding proof:
>
> > **Proposition 2 (revised)** Let $T_{1}$  and $T_2$  be two tasks such that for some state-action pair $(s_t, a_t) $, the transitions differ: $P_{1}(s_{t+1} \mid s_t, a_t)$ $\neq P_{2}(s_{t+1} \mid s_t, a_t) $. Let $z = \phi(h_{i,:t}) \in \mathcal{Z}$ be a deterministic task embedding computed from transition history. Consider two model variants:  (1) a task-state encoder, where $x_{t}=\psi(s_{t}, z)$, the transition prediction is given by $x_{t+1}\approx x_{t}+\tau(x_{t}, a_{t})$ and the estimated reward $\hat{r_{t}}=\rho(x_{t},a_{t})$; and (2) a state encoder, where $x_{t} =\psi^{\ast}(s_{t})$, $x_{t+1}\approx x_{t}+\tau^{\ast}(x_{t}, a_{t}, z)$ and the estimated reward $\hat{r_{t}}=\rho^{\ast}(x_{t},a_{t},z)$.  Then, assuming $\psi$, $\rho$ and $\tau$, as well as $\psi^{\ast}$, $\rho^{\ast}$ and $\tau^{\ast}$ are deterministic and trained to perfectly fit the transition and reward losses,
> the total entropy of the representations satisfies: $$\mathbb{H}\bigl(\psi(s_t, z)\bigr)\leq\mathbb{H}\bigl((\psi^*(s_t), z)\bigr).$$
> That is, the task-state version yields lower joint entropy due to resolving task ambiguity in the representation, resulting in simpler transitions and reward functions.
>
> > **Proof 2 (revised):**
>
> Let $s$ and $z$ be random variables representing the state and task, respectively.
> Let $\psi:\mathcal{S} \times \mathcal{Z} \to \mathcal{X}$ be a *task-aware* encoder, and
> $\psi^{*}: \mathcal{S} \to \mathcal{X}$ a *task-agnostic* encoder.
>
> Assume there exists a deterministic function $g$ such that
>
> $$\psi(s, z) = g(\psi^{*}(s), z),$$
>
> that is, the task-aware encoding can be obtained by transforming the joint variable $(\psi^*(s), z)$.
> We refer to the left-hand side as the early-context encoding (direct mapping to the abstract state-task space)
> and to the right-hand side as the late-context encoding (post-hoc adjustment within abstract state space).
>
> Since $g$ is a deterministic function applied to $(\psi^*(s), z)$, the chain rule for entropy gives:
> $$\mathbb{H}(\psi^{\ast}(s), z)=\mathbb{H}\bigl(g(\psi^{\ast}(s), z)\bigr)+\mathbb{H}\bigl((\psi^{\ast}(s), z) \big| g(\psi^{\ast}(s), z)\bigr).$$
>
> Because conditional entropy is nonnegative, we have
>
> $$\mathbb{H}\bigl(g(\psi^{\ast}(s), z)\bigr) \leq \mathbb{H}(\psi^*(s), z).$$
> Substituting $\psi(s,z) = g(\psi^{\ast}(s),z)$, we obtain
>
> $$\mathbb{H}(\psi(s, z))\leq \mathbb{H}(\psi^{\ast}(s), z).$$
>
> Equality holds if and only if $g$ is injective (i.e., information-preserving) on the support of $(\psi^{\ast}(s), z)$.
> Otherwise, any many-to-one mapping $g$ strictly reduces entropy.vHence, the early (task-state) representation cannot have higher entropy than the late (state + task) representation; it resolves task ambiguity directly in the encoded space, yielding lower representational complexity and simpler transition/reward dynamics. $\square$
>
> In the revision (Appendix), we further included a remark discussing the conditions under which strict inequality is obtained.
>
> > The baselines all seem to be other task-inference methods.
>
> - **Inclusion of other baselines:** We agree that including other baselines will strengthen our work. We refer to the Global Comments above. There we show which baselines we included and how we expanded the experimental settings.
>
> > It is unclear which methods, other than Vanilla SAC, use SAC, and it is unclear what is meant by "Vanilla PPO" and "Vanilla SAC".
>
> **Vanilla**:  Vanilla refers to regular (i.e. just PPO out-of-the-box from a library like SB3). We took this terminology from Lee et al. (2020) and will clarify the meaning in the revised version of the paper.
>
> > Less critical, but other issues that detract from a convincing paper:
>
> - **Ablation**: We refer to Figure 5, which already shows an ablation study. We will add a small ablation study on the effect of expert policy sampling (data sampled with trained PPO policy vs. random policy). Results are as follows: EMERALD PPO achieves a improvement of $214.6 \pm 129.1$ on HalfCheetah Velocity and $431.4 \pm 161.9$ on HalfCheetah Direction.
>
> - **More difficult baselines**: We included several additional experiments (using Meta-World, as you suggested). We refer to the Global Comment.

---

### Review · Reviewer_cNgU · 2025-10-06

**Summary Of Contributions:**

## Summary of Contributions

This paper proposes EMERALD, a meta-reinforcement-learning method that introduces task-aware abstract representations to improve generalization across tasks. The approach integrates task inference directly into an Abstract Representation Model (ARM) composed of a state encoder, transition model, reward predictor, and task encoder. By conditioning the state encoder on task embeddings, the model produces latent representations that are both compact and context-sensitive. I particularly appreciated the ARM framework, which cleanly separates representation learning from policy optimization — a design that feels general, reusable, and valuable beyond EMERALD itself. The authors provide theoretical justification showing that task-aware and entangled representations lead to lower loss and entropy, and demonstrate empirically that EMERALD achieves competitive results on meta-RL benchmarks.


## Strengths
	•	The methodology is coherent and well grounded. The motivation for early task-conditioning is clear, and the architecture aligns naturally with the theory.
	•	The ARM framework makes intuitive sense and provides a flexible foundation for future extensions or alternative implementations.
	•	The paper is clearly written and emphasizes modularity, making the approach compatible with standard RL algorithms such as PPO and SAC.

### Weaknesses
	•	The evaluation setup feels somewhat dated. The experiments largely follow Lee et al. (2020), focusing on CartPole, Pendulum, and HalfCheetah variants — benchmarks that are no longer particularly informative in 2025.
	•	The empirical evidence is limited in scope; while the results are positive, it is difficult to view success on these simple environments as a strong test of generalization.
	•	The set of baselines is incomplete

**Audience:**

Yes

**Audience Explanation:**

While meta-RL has become somewhat less central in recent years, there remains an active subset of researchers focused on understanding task generalization and representation learning across tasks.

**Claims And Evidence:**

No

**Claims Explanation:**

The main limitation lies in the evaluation setup, which is narrow and dated. The experimental protocol reproduces Lee et al. (2020) and relies on simple continuous-control tasks such as CartPole, Pendulum, and HalfCheetah. These benchmarks are not representative of the complexity expected in contemporary meta-RL research, and results on them cannot be taken as strong evidence of generalization in 2025.

In addition, the set of baselines is incomplete. The most recent baseline included in the paper is from 2021, which leaves out several years of progress in meta-RL and representation learning. This significantly weakens the empirical comparison, as it is unclear how EMERALD would perform against more modern architectures or methods.

Finally, the absence of a recurrent model baseline is a major omission. Prior work (e.g., Ni, Eysenbach & Salakhutdinov, 2021) has shown that a simple RNN-based policy can be surprisingly strong — often competitive with or superior to more complex meta-RL methods. Including such a baseline would provide a much fairer and more informative evaluation of EMERALD’s true benefits.

Overall, while the methodology is sound, the experimental evidence does not convincingly substantiate the paper’s claims.

**Requested Changes:**

## 1.	Modernize the evaluation setup.
The current experiments replicate Lee et al. (2020) and focus on simple benchmarks (CartPole, Pendulum, HalfCheetah), which are no longer informative or convincing. The authors should include more challenging and diverse tasks that better reflect the current state of meta-RL evaluation — for instance, Meta-World, DMControl Suite variations, or other modern continuous-control benchmarks.

Given recent trends, it could also be valuable to demonstrate the approach in LLM-based reinforcement learning. This would make the empirical section more relevant and impactful to today’s audience.

## 2.	Update and expand the baselines.
The most recent baseline currently included is from 2021. The comparison should be extended to include more recent meta-RL and representation-learning methods. In particular, the authors should add the recurrent model baseline from Ni, Eysenbach & Salakhutdinov (2021), which shows that a simple RNN trained end-to-end can be highly competitive — sometimes outperforming specialized meta-RL approaches. This would provide a much fairer and more informative test of EMERALD’s claimed benefits.

Also, the notion of task-awareness is somewhat misapplied in this work. In standard terminology, task-aware typically refers to settings where the task identity is explicitly provided to the model, rather than inferred from interaction. In EMERALD, however, the agent still operates in a task-agnostic regime — it must infer the task from experience rather than observe it directly. The terminology should therefore be revised or clarified to avoid confusion, as the current phrasing overstates the degree of task observability in the proposed setup.

---

> ### Author Response · Authors · 2025-10-19
> **Reviewer cNgU Comment**
>
> We would like to thank the reviewer for taking the time to provide valuable feedback and for the positive comments on our paper! We appreciate your suggestion on how to improve the experimental setup.
>
> We now continue addressing your concerns.
>
> # 1. Reply to Request 1 (Modernize the evaluation setup).
>
> > The current experiments replicate Lee et al. (2020) and focus on simple benchmarks (CartPole, Pendulum, HalfCheetah), which are no longer informative or convincing. The authors should include more challenging and diverse tasks that better reflect the current state of meta-RL evaluation — for instance, Meta-World, DMControl Suite variations, or other modern continuous-control benchmarks.
>
> - **Reason for Lee et al protocol:** We would first like to clarify that we follow the evaluation protocol of Lee et al. (2020) because (unlike common meta-RL benchmarks such as AntDir, HalfCheetahDir, or the Meta-World suite which primarily vary goals or task identities) it specifically tests generalization under changes in environment dynamics.
>
> - **More challenging tasks:** This is a fair point. We included more challenging tasks. In particular, we expand the current empirical by adding new results on the Meta-Worlds evaluation suite (ML1 and ML10). We refer to the Global Comment above for some results.
>
> # 2. Reply to Request 2 (Update and expand the baselines).
>
> > The most recent baseline currently included is from 2021. The comparison should be extended to include more recent meta-RL and representation-learning methods. In particular, the authors should add the recurrent model baseline from Ni, Eysenbach & Salakhutdinov (2021), which shows that a simple RNN trained end-to-end can be highly competitive — sometimes outperforming specialized meta-RL approaches. This would provide a much fairer and more informative test of EMERALD’s claimed benefits.
>
> - **Inclusion of baselines:** We do agree that it would strengthen our work to include more recent baselines and more complicated settings. The omission of recurrent baselines was also noted by Reviewer aQ3c. We included Ni, Eysenbach & Salakhutdinov (2021). We refer to the Global Comment for the results with additional baselines.
>
> > Given recent trends, it could also be valuable to demonstrate the approach in LLM-based reinforcement learning. This would make the empirical section more relevant and impactful to today’s audience.
>
> - Including LLM-based measures is certainly an interesting suggestion! However, due to the limited time span we will probably not be able to include it in this work.
>
> > Also, the notion of task-awareness is somewhat misapplied in this work. In standard terminology, task-aware typically refers to settings where the task identity is explicitly provided to the model, rather than inferred from interaction. In EMERALD, however, the agent still operates in a task-agnostic regime — it must infer the task from experience rather than observe it directly. The terminology should therefore be revised or clarified to avoid confusion, as the current phrasing overstates the degree of task observability in the proposed setup.
>
> - **Task-awareness terminology:** We understand your point, but while the task is not strictly passed to the policy, it is inferred implicitly and therefore we call it task-aware (compared to task-agnostic in which there is no awareness whatsoever). As this point was also not mentioned by the other reviewers, we prefer to keep the terminology as is.

---

### Author Response · Authors · 2025-10-19
**Global Comment to all Reviewers**

# Global Comment

We would first like to thank all the reviewers for their valuable feedback and for contributing to the improvement of our work! We were pleased to see that the reviewers appreciated:

1. The relevance of our work to the advancement of meta-RL (all reviewers).
2. The clarity and reusability of our design, as well as its general applicability (all reviewers).
3. Our theoretical motivations (cNgU and aQ3c).
4. The fairness of our comparisons and the inclusion of many relevant baselines (aQ3c).

Across the board, we identified the following valuable points for improvement:

1. Clarifying the role and necessity of Proposition 1 (aQ3c and 99cF).
2. Strengthening the rigour and clarity of Lemma 2 and Proposition 2 (aQ3c and 99cF).
3. Expanding our experimental evaluation (including updated baselines and more complex settings) (cNgU and 99cF).

We discuss each of these below and then proceed with the reviewer-specific comments.

## Role of Proposition 1:

As Reviewer 99cF pointed out, it was perhaps unclear whether Proposition 1 was needed in its original form. Reviewer aQ3c also noted that Proposition 1 did not provide new insights, as the role of a history-dependent policy is already well understood (Beck et al., 2025). We agree with these observations and have removed the proposition. Instead, following Reviewer 99cF’s suggestion, we now include a concise paragraph discussing the role of task conditioning within our framework, with references to the relevant current literature.

## Revisions to Proposition 2 and Lemmas 1 \& 2:

We recognized that Proposition 2 and Lemma 2 may have caused some confusion. To improve clarity and self-containment, we have removed Lemma's 1 and 2 and revised the proof accordingly. This revision also incorporates the reviewers' suggestions regarding the comparison of marginals versus joints (Reviewer 99cF) and the observation that a tighter bound can be obtained by comparing with $\( H(\psi^{*}(s), z) \) $(Reviewer aQ3c).

In the revised manuscript, we provide an updated version of Proposition 2, together with a more detailed and streamlined proof and a short remark on the practical implications. Updated proposition and proof are provided in our comment to Reviewer aQ3c.

## Extension of Baselines \& Modernization of Setup:

We thank all reviewers for their useful suggestions regarding the experimental setup. In particular, the suggestions from Reviewers cNgU and aQ3c were helpful in extending our experiments and strengthening the empirical support of our work.

**Additional Baselines:** As suggested, we included two more baselines, RMF-RL (Ni et al. 2021) and HN (Beck et al, 2023) within the current setup. Due to a lack of time, we could not complete all HN experiments. We will include those in the coming weeks.
|                        | CartPole (moderate) | CartPole (extreme) | Pendulum (moderate)  | Pendulum (extreme)  | HalfCheetah (moderate)            | HalfCheetah (extreme)             |
|------------------------|----------------------------|----------------------------|----------------------------|----------------------------|----------------------------|----------------------------|
| **RMF-RL (SAC)**             | 198.4 ± 2.1                | 197.2 ± 3.1                | -769.3 ± 198.1             | -1058.5 ± 238.1            | 4419.1 ± 219.3             | 2142.1 ± 323.6             |
| **HN (PPO)**              | 197.7 ± 3.1                | 194.5 ± 5.1                | -421.1 ± 43.4              | -531.3 ± 121.3             | TBD                        | TBD                        |
| **EMERALD (PPO)**        | **199.1** ± 1.1            | **199.3** ± 3.5            | -312.4 ± 19.0              | **-313.7 ± 101.5**             | 2942.5 ± 213.5             | 1401.4 ± 421.3             |
| **EMERALD (SAC)**        | 194.6 ± 2.7                | 196.2 ± 4.9                | **-198.3** ± 23.3          | -749.4 ± 340.9             | **5101.1** ± 151.8         | **3271.8** ± 180.2         |

**More complicated settings:** As suggested by Reviewers cNgU and aQ3c, in the revision we included two additional Meta-World experiments (Yu et al. 2021) on the ML1 and ML10 suites:

| Environment (ML10)  | PEARL (SAC) | RMF-RL (SAC) | VariBAD (PPO) | EMERALD (SAC) |
|--------------------|-------|-------|---------|-------------|
| **Average (Train)** | 23.1  | 46.0  | 52.1    | **53.9**    |
| **Average (Test)**  | 13.2  | 9.4   | 14.2    | **17.6**    |

| Environment (ML1) | PEARL (SAC) | RMF-RL (SAC)| VariBAD (PPO) | EMERALD (SAC) |
|----------------------------|-------|--------|---------|--------------|
| door-open-v2               | 0     | 99     | 86      | **100**      |
| basketball-v2              | 0     | **38** | 0       | 2            |
| window-open-v2             | 36    | 95     | **100** | 64           |
| pick-place-v2              | 0     | 0      | **24**  | 0            |
| button-press-topdown-v2    | 43    | 97     | 98      | **99**       |

---

> ### Comment · Reviewer_aQ3c · 2025-10-19
>
> For the sake of easy comparison, would it be possible to include the outer loop (PPO or SAC) for each method in the tables?

---

> > ### Author Response · Authors · 2025-10-19
> > **Comment to Reviewer on Additional Point**
> >
> > Certainly, we included the outer loop algorithm specifications in the tables.

---

> > > ### Comment · Reviewer_aQ3c · 2025-10-19
> > >
> > > Thank you! That’s very helpful.
> > >
> > > Also, could you add error bars to the Meta-World results and explain the choice of the error bars (e.g., standard error or confidence interval, number of seeds) in the tables?

---

> ### Author Response · Authors · 2025-10-20
> **Additional Information for Reviewer aQ3c**
>
> Sure, below are the two tables with the standard errors included (indicated with $\pm$). Unfortunately, we couldn't add them in the tables above due to the character limit. We used 3 random seeds (due to time and computational constraints we did not manage to run more for now; for the final tables in the revision our aim is to report at least 5 to 8).
>
> | Environment (ML10)  | PEARL (SAC) | RMF-RL (SAC) | VariBAD (PPO) | EMERALD (SAC) | EMERALD (PPO) |
> |--------------------|-------|-------|---------|-------------|-------------|
> | **Average (Train)** | 23.1 $\pm$ 0.4 | 46.0 $\pm$ 2.6 | 52.1 $\pm$ 5.7   | **53.9** $\pm$ 0.6   | 46.7 $\pm$ 1.85|
> | **Average (Test)**  | 13.2 $\pm$ 0.3 | 9.4 $\pm$ 1.3  | 14.2 $\pm$ 2.6   | **17.6** $\pm$ 1.0   | 15.3 $\pm$ 0.9|
>
> | Environment (ML1) | PEARL (SAC) | RMF-RL (SAC)| VariBAD (PPO) | EMERALD (SAC) |EMERALD (PPO) |
> |----------------------------|-------|--------|---------|--------------|--------------|
> | door-open-v2               | 0.0 $\pm$ 0.0    | 99  $\pm$ 0.6   | 86 $\pm $ 5.0    | **100** $\pm$ 0.0     |    92 $\pm$ 0.6        |
> | basketball-v2              | 0.0 $\pm$ 0.0     | **38** $\pm$ 1.5 | 0.0 $\pm$ 0.0        | 2.0  $\pm $ 0.5   |  4.0 $\pm$ 0.9     |
> | window-open-v2             | 36 $\pm$ 1.1   | 95   $\pm $ 1.0  | **100** $\pm $ 0.0 | 64 $\pm $ 0.4          |    65 $\pm$ 3.8     |
> | pick-place-v2              | 0.0 $\pm$ 0.0     | 0.0 $\pm$ 0.0      | **24** $\pm $ 3.6 | 0.0   $\pm $ 0.0  |     0.0 $\pm$ 0.0      |
> | button-press-topdown-v2    | 43  $\pm$ 2.5   | 97 $\pm$ 0.7    | 98    $\pm$ 1.0   | **99** $\pm $ 0.6      |     99 $\pm$ 1.0     |

---

> > ### Comment · Reviewer_aQ3c · 2025-10-20
> >
> > Thanks! And, are there results for EMERALD (PPO) on Meta-World?

---

> > > ### Author Response · Authors · 2025-10-20
> > > **Question regarding EMERALD PPO**
> > >
> > > Yes, we have currently run our method with 1 seed, and EMERALD (PPO) seems to slightly outperform VariBAD on Meta-World. To ensure a fair comparison, we plan to run 2 additional seeds in the next few days. We're happy to include the one-seed result in the table above for now, if you'd like (in that case, please let us know), and will share the full results with all 3 seeds as soon as they're available.

---

> > > > ### Comment · Reviewer_aQ3c · 2025-10-20
> > > >
> > > > Yes, please do share the results when available!

---

> > > > > ### Author Response · Authors · 2025-10-23
> > > > > **Update results**
> > > > >
> > > > > As promised, please find the results in the table above (under EMERALD (PPO)).

---

> ### Comment · Reviewer_aQ3c · 2025-10-23
>
> Thank you! I'd be very curious to see the HN experiments when available.
>
> Two questions:
> 1) Could you point me to information detailing the hyper-parameter tuning for your method and baselines?
> 2) Do you have any intuition as to why the PPO results are not as strong as the SAC results in the new experiments?

---

> > ### Author Response · Authors · 2025-10-23
> > **Reply to Additional Questions**
> >
> > Sure, we will share the results as soon as possible and let you know when they are available.
> >
> > As to your questions:
> >
> > 1. We refer to Table I in the Appendix. For the model, we used the values as described there (i.e. the ones we used for the HalfCheetah setting). For the agents (PPO and SAC), we essentially just take the standard sb3 hyperparameters and then manually adjusted the most important hyperparameters (such as the discount factor). The baselines were not specifically tuned; we used the configurations reported in their respective repositories (e.g. https://github.com/lmzintgraf/varibad) and papers (e.g. https://arxiv.org/pdf/1910.08348, table 6C).
> > 2. It seems that the weaker PPO performance likely stems from SAC’s advantage in replaying experience across multiple tasks, whereas PPO relies only on recent on-policy rollouts and thus benefits less from cross-task data in the ML10 setting.

---

> ### Comment · Reviewer_aQ3c · 2025-10-23
>
> Thank you very much for addressing my questions. The added experiments certainly clarify the pros and cons of EMERALD, and help the paper. At the moment, I still have a couple concerns about the strength of the experimental results and method of the tuning:
>
> 1) Manually tuning the hyper-parameters, but not tuning the baselines, can introduce a lot of bias toward your method -- unless you are running the exact same code on the exact environments as the baselines originally proposed. One more subtle, but still important example of this, is that which outer loop you use (PPO or SAC) is a hyperparameter. EMERALD's performance should probably be considered as an average over the two (preferably), or the best should be selected for each baseline (if there is a reason).
>
> 2) I agree that SAC can be more sample efficient than PPO, but it seems that EMERALD+PPO doesn't confer the same advantages, relative to other PPO methods, than does EMERALD+SAC, relative to other SAC methods. Unless there is a specific reason why EMERALD needs to work with PPO, it seems like the performance should be averaged, and on average, adding EMERALD does not seem to empirically help that much.
>
> 3) The empirical results seem a bit weak. The only environment that shows EMERALD is clearly better than the baselines is HalfCheetah Volume, and there, that only applies to EMERALD+SAC, which is also notably worse on HalfCheetah Direction. From the results, as a practitioner, I'd be tempted to use VariBAD PPO. Compared to EMERALD+SAC, VariBAD PPO is worse on HalfCheetah Volume, but better on HalfCheetah Direction, and more sample efficient on CartPole. Compared to EMERALD+PPO, VariBAD PPO looks better on Pendulum, HalfCheetah Volume, and maybe ML1 and ML10. I'd be especially tempted to use VariBAD, since it wasn't ever tuned for some of these environments, as far as I'm aware (e.g., HalfCheetah Volume).
>
> Overall, my concern is that the results are weak, especially given that the tuning seems to favor the proposed method.
>
> Other ambiguities remaining:
> In the paper, do you specify what the HalfCheetah Volume environment is, and what "performance" is in the tables? I assume it is the final return at the end of meta-training, averaged over a meta-episode?

---

> > ### Author Response · Authors · 2025-10-25
> > **Reply to Comment and Clarification of Concerns**
> >
> > We are happy to hear that this clarifies the pros and cons. Many thanks for the constructive feedback. To address your follow-up questions and comments:
> >
> > 1. This is a fair point. We would like to highlight two clarifications:
> >    (1) For the original experiments, we also performed manual hyperparameter tuning for the baselines (including those added during the rebuttal), though not for the new Meta-World experiments. To reduce potential bias, we are happy to rerun the two main baselines (VariBAD (PPO) and RMF-RL (SAC)) and apply exactly the same tuning process as in the original experiments.
> >    (2) Treating the outer loop as a separate hyperparameter is indeed an interesting idea. When we consider the average, we observe the following:
> >
> >    |  | CartPole (mod.) | CartPole (extr.) | Pendulum (mod.)  | Pendulum (extr.)  | HalfCheetah (mod.) | HalfCheetah (extr.) |
> >    |------------------------|---------------------|--------------------|----------------------|---------------------|-------------------------|------------------------|
> >    | **RMF-RL (SAC)**| 198.4 ± 2.1| 197.2 ± 3.1| -769.3 ± 198.1| -1058.5 ± 238.1| **4419.1 ± 219.3**| 2142.1 ± 323.6 |
> >    | **VariBAD**| **199.1 ± 2.8**| 192.0 ± 3.5| -318.5 ± 56.6| -643.1 ± 311.4| 2964.0 ± 179.5| 1618.3 ± 251.8  |
> >    | **EMERALD (avg)**| 196.9| **197.8**| **-255.4**| **-531.6** | 4021.8 | **2336.6**|
> >
> >    Overall, EMERALD achieves comparable or better average performance (except in the HalfCheetah (mod.) case). Similarly, in the new experiments, we find:
> >
> >    | Environment (ML10) | PEARL (SAC) | RMF-RL (SAC) | VariBAD (PPO) | EMERALD (SAC) | EMERALD (PPO) | **EMERALD (avg)** |
> >    |--------------------|--------------|--------------|---------------|----------------|----------------|----------------|
> >    | **Average (Train)** | 23.1 ± 0.4 | 46.0 ± 2.6 | 52.1 ± 5.7 | **53.9 ± 0.6** | 46.7 ± 1.85 | 50.3 ± 1.23 |
> >    | **Average (Test)**  | 13.2 ± 0.3 | 9.4 ± 1.3 | 14.2 ± 2.6 | **17.6 ± 1.0** | 15.3 ± 0.9 | 16.5 ± 0.95 |
> >
> >    EMERALD (avg) outperforms VariBAD on test tasks, suggesting better generalization to unseen environments.
> >
> > 2. We agree this is a relevant point and would like to provide some context. EMERALD+PPO offers clear advantages relative to vanilla PPO (e.g., HalfCheetah (volume, mod.): mean return ~800 → ~3000; Pendulum (extr.): +1000 improvement). Likewise, EMERALD+SAC outperforms RMF-RL (SAC) and SAC (vanilla), indicating that the ARM component provides a meaningful benefit. Compared to VariBAD, EMERALD PPO achieves similar performance overall (e.g., HalfCheetah Dir.: 2112 vs. 2259, effectively within margin of error). Moreover, averaging across PPO and SAC (see: two tables above), EMERALD actually seems to perform slightly better than the baselines in most cases or on par.
> >
> > 3. We appreciate the practical perspective and would like to clarify a few details:
> >    (1) EMERALD’s ARM is trained on data from a random policy, which makes the setting more challenging;
> >    (2) EMERALD’s modularity allows pairing with different algorithms, offering greater flexibility than VariBAD (typically used with PPO). Based on our results, we recommend EMERALD+SAC for practitioners;
> >   (3) Regarding the statement that “Compared to EMERALD+PPO, VariBAD PPO looks better on Pendulum, HalfCheetah Volume, and maybe ML1 and ML10,” this might be due to our not clearly stating where the baseline hyperparameter tuning was applied. As mentioned above, we did perform manual tuning for all baselines in the original experiments (including those added during the rebuttal), but not for the new Meta-World experiments. The results we refer to here should therefore be directly comparable. On Pendulum (extr.), EMERALD PPO outperforms VariBAD by ~300. In our view, on Pendulum (mod.), HalfCheetah (volume, mod.), and HalfCheetah (Direction), EMERALD PPO and VariBAD are tied. On HalfCheetah (volume, extr.), VariBAD performs slightly better than EMERALD PPO, but within the margin of error (difference ~200 with ±250).
> >
> > Overall, we believe EMERALD achieves performance on par with VariBAD (PPO), slightly better on average (PPO+SAC), and substantially better with SAC. Our focus aligns with the TMLR acceptance criteria (<https://jmlr.org/tmlr/acceptance-criteria.html>), emphasizing methodological contribution and generalization, rather than superior performance across all benchmarks. Again, we are happy to rerun the Meta-World baselines using the same tuning protocol for better comparability and methodological correctness of all experiments.
> >
> > Regarding the ambiguity: indeed, it refers to the final average over the last 100 steps at the end of meta-training (see caption of Table 1 in the manuscript).

---

> ### Comment · Reviewer_aQ3c · 2025-10-25
>
> Thanks for your ongoing work and attention to detail. A few responses:
>
> 1. (1) Yes, reproducing the same tuning process for the baselines would defintely help! (2) When I suggested the average of SAC and PPO, I meant for the baselines as well. (People have used SAC with VariBAD-based methods.) If you can't do that for the baseline, then I would only compare each EMERALD to only the baselines that use the same outer loop. I'd also probably group the tables in that way for clarity. (3) That said, the results for EMERALD Average look very similar to VariBAD to me, except on HalfCheetah. Also, isn't VariBad still better on ML1 and Cheetah Dir?
>
> 2. The most relevant baseline is a task-inference method that adds self-supervision (e.g. VariBad), and it should be compared with the same outer loop. My thoughts on VariBad are already above. The second most relevant baseline seems to be an end-to-end method. I'm curious to see the results of HN there, since that is a SOTA end-to-end method on similar tasks, and I'd also be curious to see a tuned version of RMF-RL on Meta-World (compared with the same outer loops).
>
> 3. Thanks for the clarification. In response to "On Pendulum (extr.), EMERALD PPO outperforms VariBAD by ~300. In our view, on Pendulum (mod.), HalfCheetah (volume, mod.), and HalfCheetah (Direction), EMERALD PPO and VariBAD are tied. On HalfCheetah (volume, extr.), VariBAD performs slightly better than EMERALD PPO, but within the margin of error (difference ~200 with ±250).": I'm a bit confused by the Pendulum (extr.) results. Looking at the table, you seem to be correct, but looking at the curves in the appendix, VariBAD seems to be on top. Am I missing something? Regardless, Pendulum (extr.) results look to be within the margin of error as much as the results on HalfCheetah Volume are within the margin of error, so VariBAD PPO looks very similar to EMERALD PPO.
>
> I'm but not surprised adding SAC helps, but this is not the main argument of the paper. My main takeaway from the results is that model-free methods with self-supervision (e.g. VariBAD and EMERALD) are useful, and that SAC is useful, but I'm not sure the results support the claim that EMERALD is particularly better.
>
> Regarding TMLR's criteria, I think the main question for me is whether there is a "gap between claims and evidence". For example, you claim main times to show improved performance, robustness, generalization, etc. from EMERALD. For example, it is claimed that "task-aware representation models yield substantial gains in the meta-RL setting." I'm not sure these claims are justified by the evidence.
>
> Tuning the baselines on Meta-World would be very helpful. I'm also curious to see if the results change as HN results and results with more seeds become available. Seeing SAC applied to VariBAD would also be very helpful. If that is really impossible, then it would help to at least group results by the outer-loop when presented in the paper, and not claiming superior performance/generalization for EMERALD, if it does not apply.
>
> And regarding ambiguities:
> 1. Thanks for the clarification. Over the last 100 steps of meta-training, is this the average reward in each meta-episode, or is it the final return of each meta-episode, that is averaged? Also, you may want to update the caption to clarify that this is the last 100 time steps in of meta-training (and not a meta-episode).
> 2. Do you have text that explains what the HalfCheetah Volume environment is?

---

> > ### Author Response · Authors · 2025-10-26
> > **Comment to Reply**
> >
> > >Thanks for your ongoing work and attention to detail. A few responses:
> > >
> > > 1. Yes, reproducing the same tuning process for the baselines would defintely help! (2) When I suggested the average of SAC and PPO, I meant for the baselines as well. (People have used SAC with VariBAD-based methods.) If you can't do that for the baseline, then I would only compare each EMERALD to only the baselines that use the same outer loop.
> >
> > Many thanks for all the helpful suggestions! In that case, we will rerun the baselines using the same tuning procedure and share the results as soon as they are available. We can regroup everything (i.e. include a per-outer-loop comparison) then, as you propose. To summarize, you can expect the following (in this order):
> >
> > 1. HN results.
> > 2. A rerun of the Meta-World experiments with tuned baselines.
> > 3. VariBAD + SAC results.
> >
> > If time allows, we will also include:
> >
> > 4. RMF-RL (PPO) and HN (SAC) results.
> > 5. Additional seeds.
> >
> > Items (4) and (5) will certainly be included in the final revision of the manuscript, but we cannot guarantee at this time that we will be able to complete them by the end of the rebuttal (03 Nov). If you'd like us to change the order of prioritization, please let us know.
> >
> > Regarding your last point, this is indeed correct.
> >
> > > 2. The most relevant baseline is a task-inference method that adds self-supervision (e.g. VariBad), and it should be compared with the same outer loop. My thoughts on VariBad are already above. The second most relevant baseline seems to be an end-to-end method. I'm curious to see the results of HN there, since that is a SOTA end-to-end method on similar tasks, and I'd also be curious to see a tuned version of RMF-RL on Meta-World (compared with the same outer loops).
> >
> > Sure, we will share the results as soon as possible (see our comments and list above).
> >
> > > 3. Thanks for the clarification. In response to "On Pendulum (extr.), EMERALD PPO outperforms VariBAD by ~300.
> >
> > Many thanks, this is a sharp observation. The table is correct, and it seems that something has gone wrong with the plot (likely the color assignment). We'll double-check and fix it.
> >
> > Your takeaway makes perfect sense to us. Indeed, based on the additional experiments, that seems like a fair point.
> >
> > > Regarding TMLR's criteria, I think the main question for me is whether there is a "gap between claims and evidence".
> >
> > We understand. Our reasoning behind the claim is as follows: when we compare a non-meta-RL vs. a meta-RL pair (e.g., "regular PPO" to EMERALD PPO), we do observe substantial gains. Of course, the same holds for most meta-RL approaches (e.g., "regular PPO" to VariBAD PPO). Therefore, we should add a qualifier such as *“with respect to non-meta-RL methods, task-aware representation models yield substantial gains in the meta-RL setting.”*
> >
> > It seems that your point is more about the **intra–meta-RL** comparison (e.g., VariBAD PPO vs. EMERALD PPO). On this level, the statement *“task-aware representation models yield substantial gains in the meta-RL setting”* is indeed less justified/clear, and we should weaken it accordingly (following TMLR's requirement to "for the authors to adjust (reduce) their claims.").
> >
> > Our proposal would be to rephrase the claim roughly as follows: *“Compared to standard RL approaches, we obtain substantial gains; compared to other meta-RL methods, we achieve on-par performance with VariBAD.”* (We will deliberate on the exact formulation in the coming days.) We will also ensure that this distinction is emphasized consistently throughout the manuscript. We hope this clarification better conveys where and when the claims hold.
> >
> > > Tuning the baselines on Meta-World would be very helpful. I'm also curious to see if the results change as HN results and results with more seeds become available.
> >
> > We agree. We refer to our "todo"-list above.
> >
> > > And regarding ambiguities:
> >
> > It is the final return of each meta-episode that is averaged (over the last 100 steps). We will update the caption accordingly. The HalfCheetah Volume environment is discussed in Appendix L, but we will also include a brief description in the main text, as it is currently missing.
> >
> > **As a quick sidenote:** in parallel with the experiments, we will upload a full revision (call it v2) of the manuscript sometime next week to reflect all recent updates.

---

> > > ### Comment · Reviewer_aQ3c · 2025-10-27
> > >
> > > Sounds reasonable to me. Excited to see the results!

---

> > > > ### Author Response · Authors · 2025-11-02
> > > > **Update revisions with new results**
> > > >
> > > > Great, we have just uploaded a revision of our manuscript (see: revision above). We managed to complete 1-4 and are currently working on 5.
> > > >
> > > > Please let us know if there are any remaining comments or suggestions.

---

> > > > > ### Comment · Reviewer_aQ3c · 2025-11-02
> > > > >
> > > > > Looks much better to me!
> > > > >
> > > > > One question: Is there a reason you chose VI+HN from Beck et al. (2023)? The method proposed in that paper is RNN+HN, which they show to be stronger than VI+HN.

---

> ### Author Response · Authors · 2025-11-02
> **Reply to Comment**
>
> We are happy to hear that!
>
> Regarding your question: since VI+HN is essentially VariBAD *with* a hypernetwork (i.e., a combination of an earlier approach by Beck et al. (2022), https://arxiv.org/pdf/2210.11348.pdf), we included this specific approach to improve ease of comparison, as the hypernetwork is added to a method already present in our setup (i.e., VariBAD).
>
> A second consideration was that, in the ML10 setting, RNN + HN does not necessarily seem to outperform VI +HN (we refer to https://arxiv.org/pdf/2309.14970.pdf, specifically Section 8.4 and Figure 16).
>
> If you feel that RNN+HN is a significant omission, we can include it at a later stage.

---

> > ### Comment · Reviewer_aQ3c · 2025-11-02
> >
> > Thanks for the information. I think that RNN+HN would be optimal, given the results of the 2023 paper, but if you use VI+HN, I’d at least make a note about why.

---

> > > ### Author Response · Authors · 2025-11-03
> > > **Comment to Request**
> > >
> > > Sure, for now we will make a note in the manuscript. It shouldn't be too much of a problem to include RNN + HN as well (we can do this either after or in parallel to the additional seeds experiments). We'll let you know once it's been completed.

---

### Author Response · Authors · 2025-11-16
**v3 completed**

Dear reviewers and editors,

We are pleased to share the updated version (v3) of the manuscript. We also shared the updated appendix. This revision incorporates results from eight seeds and, following Laurent’s (AE) suggestion, now includes a clearer discussion of the observed discrepancies. These points are summarized in the main text and examined in more detail in Appendix N, where we also describe the hyperparameter-tuning procedure.

Completing the full set of experiments yielded an additional insight: in the MetaWorld setting, the SAC-based variants underperform relative to the PPO outer-loop baselines. We provide an in-depth discussion of this behavior in Appendix N. Our analyses suggest that certain baselines may be less compatible with off-policy algorithms in this context. To further contextualize this observation, we conducted an extensive search for prior work pairing SAC variants (e.g., VariBAD (SAC) or RNN-HN (SAC)) with MetaWorld. So far, we have not identified previous experimental results that directly match our setting, which strengthens the relevance of highlighting this behavior.

Reviewer aQ3c noted that VariBAD has been paired with SAC before; it is therefore possible that we overlooked a relevant reference. If you are able to point us to this work, we would be grateful, and we would incorporate it into the discrepancy discussion.

Please let us know if any further revisions or clarifications are needed.

Kind regards,

The authors

---

> ### Comment · Reviewer_aQ3c · 2025-11-17
>
> Hi Authors,
>
> Thanks for your hard work in the review process! Regarding the use of SAC and VariBAD, here are a few papers that use VariBAD (or a variant thereof) with SAC. Note, several of these are in the offline setting, or have other minor architectural changes, so are not necessarily directly comparable to your results:
>
> 1. Exchangeable Models in Meta Reinforcement Learning (https://openreview.net/pdf?id=TZFlzejFPT#page13)
>
> 2. Efficient Offline Meta-Reinforcement Learning via Robust Task Representations
> and Adaptive Policy Generation (https://www.ijcai.org/proceedings/2024/0500.pdf)
>
> 3. Offline Meta Reinforcement Learning – Identifiability Challenges and Effective Data Collection Strategies (https://proceedings.neurips.cc/paper/2021/file/248024541dbda1d3fd75fe49d1a4df4d-Paper.pdf)
>
> 4. Off-Policy Meta-Reinforcement Learning With Belief-Based Task Inference (https://ieeexplore.ieee.org/ielx7/6287639/9668973/09763505.pdf?tp=&arnumber=9763505&isnumber=9668973&ref=aHR0cHM6Ly93d3cuZ29vZ2xlLmNvbS8=#page13)

---

> > ### Author Response · Authors · 2025-11-17
> > **Thanks for the paper**
> >
> > Hi Reviewer aQ3c,
> >
> > Many thanks for sharing! Despite the fact that it's perhaps not directly comparable, we are very interested in seeing how others combine VariBAD and SAC. We'll certainly have a look at the papers in the coming day.
> >
> > All the best,
> >
> > The authors

---

### Decision · Action_Editor_dMMM · 2025-11-17

**Recommendation:** Accept with minor revision

**Additional Comments:**

The authors and reviewers had a very rich discussion, and almost all of the reviewers' limitations were addressed by the authors through the discussion period. The authors also provided a few new versions of their work throughout the process (they have included an updated proof, an empirical study with additional baselines, and writing updates).

All reviewers agree that the paper can be accepted and I have checked the latest updates in the authors' latest revision (Nov. 16).

In addition, I'm sharing a few more minor suggestions:
+ Task-aware vs. task-agnostic. In the continual learning community, these terms usually indicate whether task identity is provided. Here, the authors use the term to denote something slightly different, which seems coherent with past Meta-RL work. While I don't suggest changing the terms in the paper, I would encourage the authors to add a short note early in the paper to clarify (e.g., to note that the terms are used differently across communities).
+ Similarly, in our internal discussion, a reviewer mentioned, "I believe [the term 'abstract state'] typically means there is a compressed, higher-level representation with more compression and less information than the underlying state space. However, in the paper, the abstract state instead refers to a latent or belief state in the POMDP (also referred to as a hyper-state space in some meta-RL works by Zintgraf et al), which has more information than the underlying MDP state -- not less. It could be useful to have a note of this in the paper," Again, I will leave it to the authors' discretion whether they want to make this change or not.
+ Table 4a, pick-place, shouldn't 33 be highlighted (instead of 28)?
+ The appendices are to be missing from the PDF. Also, the links to some seem to be missing.

Congratulations and many thanks for being so responsive throughout the process!

**Audience:**

Yes

**Audience Explanation:**

Meta-RL is of interest to the TMLR audience, and the reviewers found the paper to be well-executed. As such, the work will interest the community.

**Claims And Evidence:**

Yes

**Claims Explanation:**

All reviewers agree that the current version of the manuscript is correctly supported by evidence. More details below.

---

> ### Author Response · Authors · 2025-11-21
> **Thanks!**
>
> Many thanks, we are happy to hear the decision is positive! We would like to thank you and the reviewers for the time and extensive feedback and rich discussions.
>
> We are currently processing the final revisions and will share the camera-ready version of the manuscript next week.